# On Calibration and Out-of-domain Generalization

**Yoav Wald**[*]
Johns Hopkins University
yoav.wald@gmail.com

**Amir Feder**[*]
Technion
amirfeder@gmail.com

**Daniel Greenfeld**
Jether Energy Research
danielgreenfeld3@gmail.com

**Uri Shalit**
Technion
urishalit@technion.ac.il

## Abstract

Out-of-domain (OOD) generalization is a significant challenge for machine learning models. Many techniques have been proposed to overcome this challenge, often focused on learning models with certain invariance properties. In this work, we draw a link between OOD performance and model calibration, arguing that calibration across multiple domains can be viewed as a special case of an invariant representation leading to better OOD generalization. Specifically, we show that under certain conditions, models which achieve *multi-domain calibration* are provably free of spurious correlations. This leads us to propose multi-domain calibration as a measurable and trainable surrogate for the OOD performance of a classifier. We therefore introduce methods that are easy to apply and allow practitioners to improve multi-domain calibration by training or modifying an existing model, leading to better performance on unseen domains. Using four datasets from the recently proposed WILDS OOD benchmark [23], as well as the Colored MNIST dataset [21], we demonstrate that training or tuning models so they are calibrated across multiple domains leads to significantly improved performance on unseen test domains. We believe this intriguing connection between calibration and OOD generalization is promising from both a practical and theoretical point of view.

## 1 Introduction

Machine learning models have recently displayed impressive success in a plethora of fields [19, 9, 41]. However, as models are typically only trained and tested on in-domain (ID) data, they often fail to generalize to out-of-domain (OOD) data [23]. The problem is especially pressing when deploying machine learning models in the wild, where they are required to perform well under conditions that were not observed during training. For instance, a medical diagnosis system trained on patient data from a few hospitals could fail when deployed in a new hospital.

Many methods have been proposed to improve the OOD generalization of machine learning models. Specifically, there is rapidly growing interest in learning models that display certain invariance properties under distribution shifts and do not rely on spurious correlations in the training data [34, 17, 1]. While highlighting the need for learning robust models, so far these attempts have limited success scaling to realistic high-dimensional data, and in learning truly invariant representations [37, 11, 20].

In this paper, we argue that an alternative and relatively simple approach for learning invariant representations could be achieved through model calibration across multiple domains. Calibration asserts that the probabilities of outcomes predicted by a model match their true probabilities. Our claim is that simultaneous calibration over several domains can be used as an observable indicator

---

[*]Equal contribution

35th Conference on Neural Information Processing Systems (NeurIPS 2021).

for favorable performance on unseen domains. For example, if we take all patients for whom a classifier outputs a probability of $0.9$ for being ill, and in one hospital the true probability of illness in these patients is $0.85$ while in the other it is $0.95$, then we may suspect the classifier relies on spurious correlations. Intuitively, the features which lead the classifier to predict a probability of $0.9$ imply different results under different experimental conditions, suggesting that their correlation with the label is potentially unstable. Conversely, if the true probabilities in both hospitals match the classifier's output, it may be a sign of its robustness.

Our contributions are as follows: We prove that in Gaussian-linear models, under a general-position condition, being concurrently calibrated across a sufficient number of domains guarantees a model has no spurious correlations. We then introduce three methods for encouraging multi-domain calibration in practice. These are, in ascending order of complexity: (i) model selection by a multi-domain calibration score, (ii) robust isotonic regression as a post-processing tool, and (iii) directly optimizing deep nets with a multi-domain calibration objective, based on the method introduced by Kumar et al. [26]. We show that multi-domain calibration achieves the correct invariant classifier in a learning scenario presented by Kamath et al. [20], unlike the objective proposed in Invariant Risk Minimization [1]. Finally, we demonstrate that the proposed approaches lead to significant performance gains on the WILDS benchmark datasets [23], and also succeed on the colored MNIST dataset [21].

## 2   Calibration and Invariant Classifiers

### 2.1   Problem Setting

Consider observable features $X$, a label $Y$ and an environment (or domain) $E$ with sample spaces $\mathcal{X}, \mathcal{Y}, \mathcal{E}$ accordingly. We mostly focus on regression and binary classification, therefore $\mathcal{Y} = \mathbb{R}$ or $\mathcal{Y} = \{0, 1\}$. To lighten notation, our definitions will be given for the binary classification setting and we will point out adjustments to regression where necessary. There is no explicit limitation on $|\mathcal{E}|$, but we assume that training data that has been collected from a finite subset of the possible environments $E_{\text{train}} \subset \mathcal{E}$. The number of training environments is denoted by $k$, and $E_{\text{train}} = \{e_i\}_{i=1}^{k} \subset \mathcal{E}$, so that our training data is sampled from a distribution $P[X, Y \mid E = e_i] \quad \forall i \in [k]$. Our goal is to learn models that will generalize to new, unseen environments in $\mathcal{E}$.

Ideally, we would like to learn a classifier that is optimal for all environments $\mathcal{E}$. Unfortunately, we only observe data from the limited set $E_{\text{train}}$ and even if this set is extremely large, the Bayes optimal classifiers on each environment do not necessarily coincide. Following other recent work [34, 17, 1] we therefore aim for a different goal – learning classifiers whose per-instance output will be stable across environments $E$, as we explain below.

We assume the data generating process for $E, X, Y$ follows the causal graph in Figure 1. [2] We differentiate between causal and anti-causal components of $X$, and further differentiate between the anti-causal variables which are affected or unaffected by $E$, denoted as $X_{\text{ac-spurious}}$ and $X_{\text{ac-non-spurious}}$, respectively. As an illustrative example, consider again predicting illness across different hospitals. When predicting lung cancer, $Y$, from patient health records, $X_{\text{causal}}$ could be features like smoking. $X_{\text{ac-non-spurious}}$ are symptoms of $Y$ such as infections that appear in chest X-rays, while $X_{\text{ac-spurious}}$ can be marks that technicians put on X-rays as in [51]. Smoking habits may vary across hospital populations, as might X-ray markings; but the influence of smoking on cancer and the manifestation of cancer in an X-ray do not vary by hospital.

We do not assume to know how to partition $X$ into $X_{\text{causal}}, X_{\text{ac-spurious}}, X_{\text{ac-non-spurious}}$. The main assumptions made in the causal graph in Fig. 1 are that there are no hidden variables, and that there is no edge directly from environment $E$ to the label $Y$. Such an arrow would imply the conditional distribution of $Y$ given $X$ can be arbitrarily different in an unseen environment $E$, compared to those present in the training set. Note that for simplicity we do not include arrows from $X_{\text{causal}}$ to $X_{\text{ac-spurious}}$ and $X_{\text{ac-non-spurious}}$ but they may be included as well.

Figure 1: Learning in the presence of causal and anti-causal features. Anti-causal features can be either spurious ($X_{\text{ac-spurious}}$), or non-spurious ($X_{\text{ac-non-spurious}}$).

---

[2]See Appendix A.3 for a brief introduction to causal graphs.

We will say a representation $\Phi(X)$ contains a *spurious correlation* with respect to the environments $E$ and label $Y$, if $Y \not\perp\!\!\!\perp E \mid \Phi(X)$; this motivates our naming of $X_{\text{ac-spurious}}$ and $X_{\text{ac-non-spurious}}$ in Fig. 1, as $Y \not\perp\!\!\!\perp E \mid X_{\text{ac-spurious}}$ but $Y \perp\!\!\!\perp E \mid X_{\text{ac-non-spurious}}$. Similar observations have been made by [17, 1]. Having a spurious correlation implies that the relation between $\Phi(X)$ and $Y$ depends on the environment – it is not transferable nor stable across environments. In this work we will simply consider the output $f(X)$ of a classifier $f : \mathcal{X} \to [0, 1]$ as a representation. The crux of this paper is the observation that having $\mathbb{E}[Y \mid f(X), E = e] = f(X)$ for every value of $E$, i.e. $f$ being a ***calibrated*** classifier across all environments, is equivalent up-to a simple transformation to having $Y \perp\!\!\!\perp E \mid f(X)$, and thus to $f$ having ***no spurious correlations*** with respect to $E$. We prove this assertion in section 2.2, and as a demonstration of this principle we prove (section 3) that linear models which are calibrated across a diverse set of environments $E$ are guaranteed to discard $X_{\text{ac-spurious}}$ as viable features for prediction.

## 2.2 Invariance and Calibration on Multiple Domains

We define calibration, along with a straightforward generalization to the multiple environment setting.

**Definition 1.** *Let $P[X, Y]$ be a joint distribution over the features and label, and $f : \mathcal{X} \to [0, 1]$ a classifier. Then $f(\mathbf{x})$ is calibrated w.r.t to $P$ if for all $\alpha \in [0, 1]$ in the range of $f$, $\mathbb{E}_P[Y \mid f(X) = \alpha] = \alpha$. In the multiple environments setting, $f(\mathbf{x})$ is calibrated on $E_{train}$ if for all $e_i \in E_{train}$ and $\alpha$ in the range of $f$ restricted to $e_i$, $\mathbb{E}[Y \mid f(X) = \alpha, E = e_i] = \alpha$.*

For regression problems, we consider regressors that output estimates for the mean and variance of $Y$, and say they are calibrated if they match the true values similarly to the definition above. The precise definition can be found in the supplementary material.

We now tie the notion of calibration on multiple environments with OOD generalization, starting with its correspondence with our definition of spurious correlations. Recall that a representation $\Phi(X)$ does not contain spurious correlations if $Y \perp\!\!\!\perp E \mid \Phi(X)$. Treating the output $f(X)$ of a classifier as a representation of the data, and considering classifiers satisfying the above conditional independence with respect to training environments, we arrive at a definition of an invariant classifier.

**Definition 2.** *Let $f : \mathcal{X} \to [0, 1]$. $f$ is an* invariant classifier *w.r.t $E_{train}$ if for all $\alpha \in [0, 1]$ and environments $e_i, e_j \in E_{train}$, where $\alpha$ is in the range of $f$ restricted to each of them:*

$$\mathbb{E}[Y \mid f(X) = \alpha, E = e_i] = \mathbb{E}[Y \mid f(X) = \alpha, E = e_j]. \tag{1}$$

Lemma 1 gives the correspondence between invariant classifiers and classifiers calibrated on multiple environments. The proof is in Section A.1 of the supplementary material.

**Lemma 1.** *If a binary classifier $f$ is invariant w.r.t $E_{train}$, then there exists some $g : \mathbb{R} \to [0, 1]$ such that (i) $g \circ f$ is calibrated on all training environments, and (ii) the mean squared error of $g \circ f$ on each environment does not exceed that of $f$. On the other hand, if a classifier is calibrated on all training environments it is also invariant w.r.t $E_{train}$.*

Now, we can note how the above notion of invariance relates to that of Invariant Risk Minimization [1], where invariance of a representation $\Phi : \mathcal{X} \to \mathcal{H}$ is linked to a shared classifier $\mathbf{w}^* : \mathcal{H} \to [0, 1]$, $\mathbf{w}^* \circ \Phi$ being optimal on all environments w.r.t a loss $l : [0, 1] \times \mathcal{Y} \to \mathbb{R}_{\geq 0}$. Under the representation $\Phi(X) = f(X)$, and the cross-entropy or squared losses it turns out that the original IRM definition coincides with Equation (1) [3]. Hence we aim for a similar notion of conditional independence, yet we approach it from the point-of-view of calibration. In Section 5 we will see that taking this approach leads to different methods that are highly effective in achieving and assessing invariance. We further note that the original IRM objective was deemed too difficult to optimize by the original IRM authors, leading them to propose an alternative called IRMv1. This alternative however does not capture the full set of required invariances, as shown by [20], whereas we show in section 6.1 that multi-domain calibration does indeed capture the required invariances.

Having established the connection between calibration on multiple environments and invariance, there are several interesting questions and points to consider:
**Calibration and sharpness.** Calibration alone is not enough to guarantee that a classifier performs well; on a single environment, always predicting $\mathbb{E}[Y]$ will give a perfectly calibrated classifier.

---

[3]See Observation 2 in [20] for a proof.

Hence, multi-domain calibration should be combined with some sort of guarantee on accuracy. In the calibration literature, this is often referred to as sharpness. To this end, in Section 5 we will propose regularizing models during training or fine-tuning with Calibration Loss Over Environments (CLOvE). Combining this regularizer with standard empirical loss functions helps balance between sharpness and multi-domain calibration. Even without training a new model, we will propose methods for model selection and post-processing that are very easy to apply and help improve multi-domain calibration without a significant effect on the sharpness of the models.

**Generalization and dependence on $X_{\text{ac-spurious}}$.** Suppose that $f(X)$ is calibrated on $E_{\text{train}}$. Under what conditions does this imply it is calibrated on $\mathcal{E}$? It is easy to show that calibration on several environments entails calibration on any distribution which can be expressed as a linear combination of the distributions underlying said environments. However, can we go beyond that? Given a general set $\mathcal{E}$ we would like to know what conditions and how many training environments are required for calibration to generalize. We also wish to understand when does calibration over a finite set of training environments indeed guarantee that a classifier is free of spurious correlations. We now turn to answer these questions in the setting of linear-Gaussian models.

## 3   Motivation: a Linear-Gaussian Model

Let us consider data where $X$ is a multivariate Gaussian. Since we will be considering Gaussian data, the set of all environments $\mathcal{E}$ will be parameterized using pairs of real vectors expressing expectations and positive definite matrices of an appropriate dimension expressing covariances: $\mathcal{E} = \{(\mu, \Sigma) \mid \mu \in \mathbb{R}^d, \Sigma \in \mathbb{S}^d_{++}\}$.

For two scenarios ((a) and (b) in Figure 2) we prove that when provided with data from $k$ training environments, where $k$ is linear in the number of features, and the environments satisfy some mild non-degeneracy conditions, any predictor that is calibrated on all training environments will not rely on any of the spurious features $X_{\text{ac-sp}}$, and will also be calibrated on all $e \in \mathcal{E}$.

In scenario (a), we take $Y$ to be a binary variable drawn from a Bernoulli distribution with parameter $\eta \in [0, 1]$, and observed features are generated conditionally on $Y$. The features $\mathbf{x}_{\text{ac-ns}} \in \mathbb{R}^{d_{\text{ns}}}$ are invariant, meaning their conditional distribution given $Y$ is the same for all environments, whereas $\mathbf{x}_{\text{ac-sp}} \in \mathbb{R}^{d_{\text{sp}}}$ are spurious features, as their distribution may shift between environments, altering their correlation with $Y$. The data generating process for training environment $i \in [k]$ in Fig. 2(a) is given by:

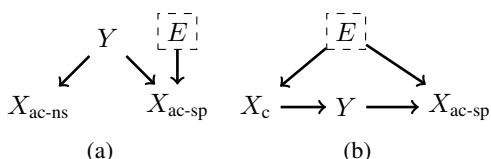

(a)                              (b)

Figure 2: Graphs describing our theoretical analysis. Features are: (a) anti-causal, some spurious while others invariant. (b) causal and may undergo covariate shift, or anti-causal and spurious.

$$y = \begin{cases} 1 & \text{w.p } \eta \\ 0 & \text{o.w} \end{cases} \qquad \begin{aligned} X_{\text{ac-ns}} \mid Y = y &\sim \mathcal{N}\left((y - 1/2)\mu_{\text{ns}}, \Sigma_{\text{ns}}\right), \\ X_{\text{ac-sp}} \mid Y = y &\sim \mathcal{N}\left((y - 1/2)\mu_i, \Sigma_i\right). \end{aligned} \qquad (2)$$

For $\mathbf{x} = [\mathbf{x}_{\text{ac-ns}}, \mathbf{x}_{\text{ac-sp}}]$ we consider a linear classifier $f(\mathbf{x}; \mathbf{w}, b) = \sigma(\mathbf{w}^\top \mathbf{x} + b)$, where $\sigma : \mathbb{R} \to [0, 1]$ is some invertible function (e.g. a sigmoid). Since the mean of spurious features, $\mu_i$, is determined by $y$, these features can help predict the label in some environments. Yet, these correlations do not carry to all environments, and $f(\mathbf{x})$ might rely on spurious correlations whenever the coefficients in $\mathbf{w}$ corresponding to $\mathbf{x}_{\text{ac-sp}}$ are non-zero. Any such classifier can suffer an arbitrarily high loss in an unseen environment, because a new environment can reverse and magnify the correlations observed in $E_{\text{train}}$. Using these definitions, we may now state our result for this case:

**Theorem 1.** *Given $k > 2d_{sp}$ training environments where data is generated according to Equation (2) with parameters $\{\mu_i, \Sigma_i\}_{i=1}^k$, we say they lie in general position if for all non-zero $\mathbf{x} \in \mathbb{R}^{d_{sp}}$:*

$$\dim\left(\text{span}\left\{\begin{bmatrix} \Sigma_i \mathbf{x} + \mu_i \\ 1 \end{bmatrix}\right\}_{i \in [k]}\right) = d_{sp} + 1.$$

*If a linear classifier is calibrated on $k$ training environments which lie in general position, then its coefficients for the features $\mathbf{x}_{ac\text{-}sp}$ are zero. Moreover, the set of training environments that do not lie in general position has measure zero in the set of all possible training environments $\mathcal{E}^k$.*

As a corollary, we see that calibration on training environments generalizes to calibration on $\mathcal{E}$. The proof of this theorem is given in the supplementary material, Section A.4. The data generating process closely resembles the one considered by [37], who use diagonal covariance matrices.

In the second scenario we consider the addition of causal features subject to covariate shift $\mathbf{x}_c \in \mathbb{R}^{d_c}$, as shown in Figure 2b. The covariate shift is induced when the environments $E$ alter the distribution of the causal features $\mathbf{x}_c$ [40]. In this case, we analyze a regression problem since it is amenable to exact analysis. The data generating process for training environment $i \in [k]$ is:

$$X_c \sim \mathcal{N}(\mu_i^c, \Sigma_i^c); \ Y = \mathbf{w}_c^{*\top} \mathbf{x}_c + \xi, \ \xi \sim \mathcal{N}(0, \sigma_y^2)$$
$$X_{\text{ac-sp}} = y\mu_i + \eta, \ \eta \sim \mathcal{N}(\mathbf{0}, \Sigma_i). \tag{3}$$

For $\mathbf{x} = [\mathbf{x}_c, \mathbf{x}_{\text{ac-sp}}]$ it turns out that in this case, calibration on multiple domains forces $f(\mathbf{x})$ to discard $\mathbf{x}_{\text{ac-sp}}$, but also forces it to use $\mathbf{w}_c^*$, since it characterizes $P(Y \mid \mathbf{x}_c)$ which is the invariant mechanism in this scenario. The exact statement and proof are in Section A.5 of the supplement.

**Theorem 2** (informal). *Let $f(\mathbf{x}; \mathbf{w}) = \mathbf{w}^\top \mathbf{x}$ be a linear regressor and assume we have $k > \max\{d_c + 2, d_{sp}\}$ training environments where data is generated according to Equation (3). Under mild non-degeneracy conditions, if the regressor is calibrated across all training environments then the coefficients corresponding to $X_c$ equal $\mathbf{w}_c^*$ and those that correspond to $X_{ac\text{-}sp}$ are zero.*

Together, these results show calibration can generalize across environments, given that the number of environments is approximately that of the spurious features. They also show that for the settings above, the relatively stable and well-known notion of calibration implies avoiding spurious correlations.

## 4 Related Work

As discussed in Section 2, multi-domain calibration is an instance of an invariant representation [1]. Many extensions to the above work have been proposed, e.g. [24, 3]. Yet, recent work claims that many of these approaches still fail to find invariant relations in cases of interest [20, 37, 13], where a significant challenge seems to be the gap between what is achieved by the regularization term used in practice and the goal of conditional independence $Y \perp\!\!\!\perp E \mid \Phi(X)$. Gulrajani et al. [11] give a sobering view on methods for OOD generalization, emphasizing the power of ERM and data augmentation, and the challenge of model selection. We claim that compared to the above approaches, multi-domain calibration studied here is a simpler form of invariance. Furthermore, calibration is attractive because there are standard tools to quantify it such as calibration scores [31] and a vast literature on its properties and how it can be obtained [50, 47, 30, 26, 45, 14, 36].

Learning models which generalize OOD is a fruitful area of research with many recent developments. Most work focuses on the case of Domain Adaptation where unlabeled samples are available from the target domain, including recent work on OOD calibration [48]. However, important work has also been done on the area of our focus – the so-called "proactive" case [43], where no OOD samples are available whatsoever [28, 17, 38, 34, 39].

Calibration also plays an important role in uncertainty estimation for deep networks [12], and recently in fairness, where calibration on subgroups of populations is sought [35]. This has interesting resemblance to the multiple environments calibration we consider here. A more general notion of multi-calibration has also been studied in this context [16], with recent results on sample complexity [42] which may provide tools to finite sample analysis of domain generalization. Finally, multiple methods for training calibrated models [26, 29, 36] have also been proposed. In Section 5 we propose a generalization of [26] to the multi-domain case to achieve multi-domain calibration.

## 5 Proactively Achieving Multi-Domain Calibration

So far we have seen a general argument why calibration can limit spurious correlations, and that in linear-Gaussian models multi-domain calibration guarantees OOD generalization. Now we turn to a more applied perspective and show how can we optimize models so they achieve this type of calibration in practice. We propose three approaches: (1) using calibration measures for model selection, (2) post-processing calibration, and (3) a calibration objective building on a method proposed by [26]. Section A.1 in the supplementary provides a slightly broader introduction to notions we use here. We will assess model calibration by the Expected Calibration Error (*ECE*) of the calibration curve [7], which is the average deviation between model accuracy and model confidence.

## 5.1 Model selection with average ECE

Model selection is challenging when aimed at OOD generalization. As recently observed by [11], since OOD accuracy is often at odds with In-Domain (ID) accuracy, selection based on ID validation error eliminates the advantage of domain generalization methods over vanilla ERM with data augmentation. We suggest that model selection towards OOD generalization should balance ID validation error with another observable surrogate for the stability of a model to distribution shifts between domains. Motivated by multi-domain calibration, we propose using the average ECE across training environments as this surrogate. Concretely, we propose choosing a model with lowest average ECE from those obtaining ID validation accuracy that is above a certain user-defined threshold.

## 5.2 Post-Processing Calibration

Practitioners interested in (single-domain) calibrated models often apply post-processing calibration methods to binary classifiers, where the most widely used approach is Isotonic Regression Scaling [50, 30]. Unlike standard calibration problems, in our case there are multiple domains to calibrate over. We give two ways of extending Isotonic Regression to the multi-domain setting, which we term "naive calibration" and "robust calibration". **Naive Calibration** takes predictions of a trained model $f$ on validation data pooled from all domains and fits an isotonic regression $z^*$. We then report the performance of $z^* \circ f$ on the OOD test set.
**Robust Calibration:** In a multiple domain setting, Naive calibration may produce a model that is well calibrated on the pooled data, but uncalibrated on individual environments. Since our goal is simultaneous calibration, the following alternative attempts to bound the worst-case miscalibration across training environments. For each environment $e \in E_{\text{train}}$, we denote the number of validation examples we have from it by $N_e$, and by $f_{e,i}$ the prediction of the model on the $i$-th example. Then in a similar vein to robust optimization, we fit an isotonic regressor that

solves: $z^* = \arg \min_z \max_{e \in E_{\text{train}}} \frac{1}{N_e} \sum_{i=1}^{N_e} \left( z(f_{e,i}) - y_i \right)^2$. Since Isotonic Regression can be formulated as

a quadratic program, and Equation (5.2) minimizes a pointwise maximum over such objectives, we can cast Eq. 5.2 as a convex program and solve with standard optimizers. We then evaluate the OOD performance of $z^* \circ f$.

## 5.3 Learning with Multi-Domain Calibration Error

The above model selection and post-processing methods are easy to apply and (as we will soon see) surprisingly effective. However, both are limited in their power to learn a model that is truly well-calibrated across multiple domains. We now propose a more powerful approach: an objective function that directly penalizes calibration errors on multiple domains during training. Specifically, we propose learning a parameterized classifier $f_\theta(\mathbf{x})$ using a learning rule of the form: $\min_\theta \sum_{e \in E_{\text{train}}} l^e(f_\theta) + \lambda \cdot r(f_\theta)$, where $l : \mathbb{R} \times \mathbb{R} \to \mathbb{R}$ is an empirical loss function (e.g. cross-entropy) and $l^e(f_\theta)$ denotes the expected loss over data from training environment $e$, and $r(f_\theta)$ is a regularization term over multiple environments. Using this notation the method proposed by [1] learns a classifier $f = w \circ \Phi$ with a regularizer given by $r(f) = \sum_{e \in E_{\text{train}}} r_{\text{IRMv1}}^e(f)$, where $r_{\text{IRMv1}}^e(f) = \|\nabla_{w|w=1} l^e(w \cdot \Phi)\|^2$.

Our proposed regularizer $r(f_\theta)$ is based on the work of Kumar et al. [26], who introduce a method they call Maximum Mean Calibration Error (MMCE). MMCE harnesses the power of universal kernels to express the ECE as an Integral Probability Measure, and works as follows: For a dataset $D = \{\mathbf{x}_i, y_i\}_{i=1}^m$, denote the confidence of a classifier on the $i$-th example by $f_{\theta;i} = \max\{f_\theta(x_i), 1 - f_\theta(x_i)\}$ and its correctness by $c_i = \mathbb{1}_{|y_i - f_{\theta;i}| < \frac{1}{2}}$. For a given universal kernel $k : \mathbb{R} \times \mathbb{R} \to \mathbb{R}$, MMCE over the dataset $D$ is given by: $r_{\text{MMCE}}^D(f_\theta) = \frac{1}{m^2} \sum_{i,j \in D} (c_i - f_{\theta;i})(c_j - f_{\theta;j}) k(f_{\theta;i}, f_{\theta;j})$.
**Calibration Loss Over Environments (CLOvE).** Given multiple training domains with a dataset $D^e$ for each $e \in E_{\text{train}}$, we arrive at our proposed regularizer by aggregating MMCE over them: $r_{\text{CLOvE}}(f_\theta) = \sum_{e \in E_{\text{train}}} r_{\text{MMCE}}^{D_e}(f_\theta)$. A key property of CLOvE is that its minima correspond to perfectly calibrated classifiers over all training domains, a consequence of the correspondence between MMCE and perfect calibration.

**Corollary 1** (of Thm. 1 in [26]). *CLOvE is a proper scoring rule. That is, it equals $0$ if and only if $f_\theta(\mathbf{x})$ is perfectly calibrated for every $e \in E_{train}$.*

Additional properties of CLOvE, such as large deviation bounds and relation to ECE, can also be derived; see results in [26] for further details. In the following section, we will see how these properties translate into favorable OOD generalization in practice when training with CLOvE.

## 6 Experiments and Results

### 6.1 Colored MNIST and Two-Bit Environments

In order to explore the challenges of OOD generalization and how they relate to learning from multiple environments, [1] used the colored MNIST dataset [21]. In this dataset certain digits tend to be colored either red or green in the train set, but the correlation between colors and digits is flipped in the OOD test set, making color a spurious feature. This dataset was then further simplified into "Two-Bit" environments by [20], who proved that the IRMv1 penalty proposed in [1] does *not* in fact achieve the correct invariant solution on the simplified setting. The Two-Bit environments problem setting has two binary features, $X_1, X_2 \in \{-1, 1\}$, corresponding respectively to digit identity $(0 - 4$ or $5 - 9)$ and digit color in the original colored MNIST. The environments are parameterized by $e = (\alpha, \beta) \in [0, 1]^2$ controlling the correlation of the features with the label:
$Y \leftarrow \mathrm{Rad}(0.5), \ X_1 \leftarrow Y \cdot \mathrm{Rad}(\alpha), X_2 \leftarrow Y \cdot \mathrm{Rad}(\beta)$, where $\mathrm{Rad}(\delta)$ is a random variable equal to $-1$ with probability $\delta$ and 1 with probability $1 - \delta$. At training we are given data from two environments $e_1 = (\alpha, \beta_1), e_2 = (\alpha, \beta_2), \beta_1 \neq \beta_2$. The learned model is tested on a new environment $e_3 = (\alpha, \beta_3)$ with $\beta_3$ significantly different from $\beta_1, \beta_2$. Only a model discarding the spurious feature $X_2$ will maintain its accuracy moving from train to OOD test.

**Calibration discards spurious correlation in Two-Bit environments.** Figure 3(a), which we adapt from Figure 6 in Appendix B of [20], illustrates the merits of CLOvE in this setting. The figure shows the space of odd classifiers, i.e. those for which $f(1, -1) = -f(-1, 1)$, and $f(1, 1) = -f(-1, -1)$.[4] The true invariant classifiers are those for which in addition $f(1, 1) = f(1, -1)$, corresponding to models lying on the diagonal of Figure 3(a), denoted by the dashed gray line. In the figure, we plot in solid lines the classifiers for which $r_{\mathrm{IRMv1}}^e(f)$ equals 0, and in solid circles the classifiers for which $r_{\mathrm{MMCE}}^e(f)$ equals 0 (due to Corollary 1 these coincide with calibrated classifiers on environment $e$). Note that in this parameterization, the zeros of $r_{\mathrm{IRMv1}}^e(f)$ are lines whereas the zeros of $r_{\mathrm{MMCE}}^e(f)$ are isolated points. Intersections of the zeros of $r_{\mathrm{IRMv1}}^e(f)$ denote solutions for which the corresponding regularization terms are 0 on all respective environments, while intersection of zeros of $r_{\mathrm{MMCE}}^e(f)$ are the zeros of $r_{\mathrm{CLOvE}}(f)$. As observed by [20], when $E_{\mathrm{train}} = \{e_1, e_2\}$ the solution denoted by $\mathrm{OPT}_{\mathrm{IRMv1}}$ has the lowest empirical loss, yet this solution has a spurious correlation with $X_2$ and thus will incur a higher loss on the test environment $e_3$. This means the corresponding IRMv1 learning rule cannot retrieve the optimal invariant classifier. On the other hand, learning with CLOvE does retrieve the optimal invariant classifier in this case, in addition to the trivial, constant classifier. This means CLOvE discards spurious correlations in cases where IRMv1 does not. In Section C we present experiments reproducing the above scenario on the Colored MNIST dataset.

**Model selection based on average ECE** We train models with varying hyperparameters on Colored MNIST using ERM, CLOvE and IRM, (100 models with each algorithm, see Section C of the supplement for details). We then calculate the ECE and IRMv1 penalties of each model over a held-out validation set from each training environment, and evaluate the average of these against OOD accuracy. Figure 3(b) presents the results across all trained models. The ID ECE penalty displays a very strong correlation across the entire range and every training regime (Pearson corr. = -0.92), while ID IRMv1 behaves more erratically (Pearson corr. = -0.59). Since quantities used for model selection should be agnostic to choices made at training time, we suggest that ID ECE is a better choice for use in model selection. Further results on model selection can be found in the supplement, Section C.

### 6.2 WILDS Benchmarks

WILDS is a recently proposed benchmark of in-the-wild distribution shifts from several data modalities and applications[5]. Table 1 presents the four WILDS datasets we experiment with, chosen to represent diverse OOD generalization scenarios. We follow the models and training algorithms

---

[4]As explained in [20], the optimal solutions are odd so we may focus on them for visualization purposes.
[5]https://wilds.stanford.edu

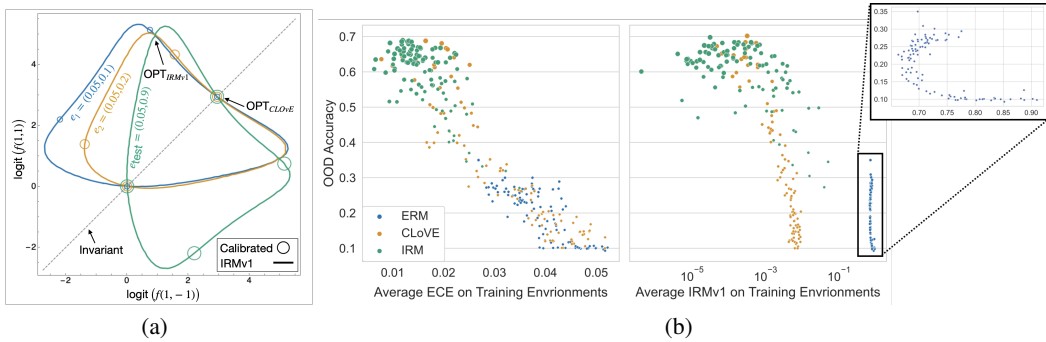

(a)                                                                          (b)

Figure 3: (a) Zeros of MMCE and IRMv1 are indicated by circles and by solid lines respectively, in a color corresponding to each environment. The dashed diagonal is the space of invariant solutions. Some zeros intersect across environments, and these are therefore the domain-invariant solutions. Among the domain-invariant solutions, $OPT_{IRMv1}$ has the lowest empirical loss when training on $e_1, e_2$. Hence learning with IRMv1 will prefer this model over $OPT_{CLOvE}$, which discards the spurious correlation with $X_2$. (b) Correspondence between observable criteria and OOD accuracy in CMNIST. Each point corresponds to a model trained with some training algorithm (marked by color) and hyperparameter setting. Size of marker is proportional to the ratio between OOD and ID accuracies.

proposed by [23]. In order to perform multi-domain calibration we modify the splits to include a multi-domain validation set whenever possible. See supplemental Section B for details and for additional results on Amazon Reviews. As in [23], we use three different training algorithms to train our models: **ERM**, **IRM**, **DeepCORAL**, and further use **GroupDRO** for one of the datasets, compatible with WILDS version 1.0.0. We apply three calibration approaches described in 5.2 and 5.3 above to each trained model: **naive calibration** and **robust calibration**, which are post-processing methods and therefore applied on the models' outputs; and **CLOvE**, which we apply as a fine-tuning approach to the top layers of each trained model. We train each (algorithm $\times$ calibration) combination four times with different random seeds, and report average results and their standard deviations.

| Dataset | Type | Label ($y$) | Input ($x$) | Domain ($e$) | Model ($f(x)$) |
|---|---|---|---|---|---|
| **PovertyMap** | Regression | Asset Wealth Index | Satellite Image | Country | ResNet |
| **Camelyon17** | Binary | Tumor Tissue | Histopathological Image | Hospital | DenseNet |
| **CivilComments** | Binary | Comment Toxicity | Online Comment | Demographics | BERT |
| **FMoW** | Multi-class | Land Use Type | Satellite Image | Region | DenseNet |

Table 1: Description of each of the datasets used in our WILDS experiments.

Table 2 presents our main results on the *FMoW* (left) and *Camelyon17* (right) datasets. On both datasets, robust calibration already improves performance, and CLOvE then significantly outperforms robust calibration, improving performance by 7% and 2.8% (absolute) over the strongest alternative on *FMoW* and *Camelyon17*, respectively. When compared to the original model, the performance of CLOvE is even more striking, with CLOvE outperforming it by more than 10% (absolute) on *FMoW* and 6% on *Camelyon17*. Another appealing property of CLOvE is the low variance exhibited across different runs. Indeed, CLOvE has lower variance than both naive and robust calibration approaches, and has lower variance than the original (uncalibrated) model on 4 of the 6 experiments.

| | *FMoW* | | | | *Camelyon17* | | | |
|---|---|---|---|---|---|---|---|---|
| Algorithm | Orig. | Naive Cal. | Rob. Cal. | CLOvE | Orig. | Naive Cal. | Rob. Cal. | CLOvE |
| ERM | 32.63 | 33.09 | 37.19 | **44.16** | 66.66 | 71.23 | 71.22 | **75.75** |
| | (0.016) | (0.021) | (0.035) | (0.018) | (0.144) | (0.089) | (0.086) | (0.049) |
| DeepCORAL | 31.73 | 31.75 | 33.86 | **40.05** | 72.44 | 75.97 | 76.8 | **79.96** |
| | (0.01) | (0.01) | (0.016) | (0.009) | (0.044) | (0.054) | (0.065) | (0.039) |
| IRM | 31.33 | 31.81 | 34.41 | **42.24** | 70.87 | 73.25 | 73.4 | **73.95** |
| | (0.012) | (0.016) | (0.015) | (0.014) | (0.068) | (0.066) | (0.069) | (0.061) |

Table 2: Left: worst unseen region accuracy on OOD test set in *FMoW*. Right: Accuracy on unseen hospital test set in *Camelyon17*. Orig.: original algorithm, no changes applied. Best OOD result for each domain in **bold**. Standard deviation across runs in brackets, lowest OOD std. is underlined.

**Analysis.** As can be seen in Figure 4, improvements in ID calibration are associated with better OOD performance. Interestingly, when our post-processing does not improve OOD performance, it

is often linked to our inability to substantially improve ID calibration. This is most visible in IRM experiments, where robust calibration is unable to outperform naive calibration both in terms ID calibration and in OOD performance. Finally, we find it interesting that merely post-processing the data (as in robust calibration) can already have such a marked effect on OOD accuracy, though still inferior to actually optimizing for multi-domain calibration as done by CLOvE.

**Results on alternative settings.** While our theoretical analysis is focused on OOD generalization of classification models, we also experiment with alternative settings from *WILDS* to test the power of ID calibration in improving OOD performance. Specifically, we experiment with the *PovertyMap* dataset, which introduces a regression task, and the *CivilComments* dataset, which introduces a sub-population shift scenario for a binary classifier. As can be seen in Table 3, results on the *CivilComments* dataset (right), show that calibration consistently improves worst-case performance, with an average improvement of $21.5\%$ across training algorithms. While CLOvE does outperform naive and robust calibration on average, the gain is lower in comparison to *FMoW* and *Camelyon17*.

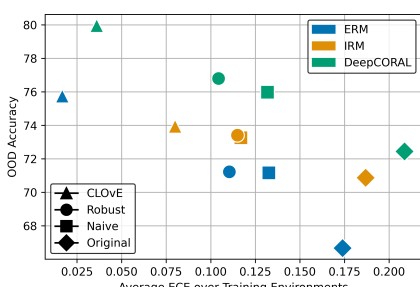

Figure 4: OOD accuracy as a function of average ECE over training domains, for all models on the *Camelyon17* dataset.

In *PovertyMap* (left), the model solves a regression task, so we cannot use CLOvE to improve OOD performance. Still, robust calibration improves performance across all experiments, though by a smaller margin. In the case of models pre-trained by IRM, robust calibration improves OOD performance substantially, outperforming the original model by $0.08\%$ (absolute). Interestingly, calibration also leads to more stable results both in *PovertyMap* and in *CivilComments*, as can be seen in the standard deviation across different model runs.

| | *PovertyMap* | | | | *CivilComments* | | | |
|---|---|---|---|---|---|---|---|---|
| Algorithm | Orig. | Naive Cal. | Rob. Cal. | Algorithm | Orig. | Naive Cal. | Rob. Cal. | CLOvE |
| ERM | 0.832 | 0.827 | **0.834** | ERM | 63.65 | 76.98 | 78.99 | **80.39** |
| | (0.011) | (0.014) | (0.006) | | (0.026) | (0.005) | (0.008) | (0.007) |
| IRM | 0.735 | 0.812 | **0.815** | IRM | 40.61 | **68.97** | 68.92 | 68.45 |
| | (0.117) | (0.016) | (0.015) | | (0.16) | (0.013) | (0.013) | (0.02) |
| DeepCORAL | 0.832 | 0.835 | **0.837** | GroupDRO | 71.67 | 76.2 | 78.54 | **80.07** |
| | (0.011) | (0.009) | (0.012) | | (0.007) | (0.013) | (0.008) | (0.003) |

Table 3: Left: Pearson correlation $r$ on in-domain (ID) and OOD (unseen countries) test sets in *PovertyMap*. Right: worst-case group accuracy on the test set in the *CivilComments* dataset.

# 7 Conclusion

In this paper we highlight a novel connection between multi-domain calibration and OOD generalization, arguing that such calibration can be viewed as an invariant representation. We proved in a linear setting that models calibrated on multiple domains are free of spurious correlations and therefore generalize out of domain. We then proposed multi-domain calibration as a practical and measurable surrogate for the OOD performance of a classifier. We demonstrated that actively tuning models to achieve multi-domain calibration significantly improves model performance on unseen test domains, and that in-domain calibration on a validation set is a useful criterion for model selection. A major limitation of our work is that our theoretical findings are limited to linear models in a population (as opposed to finite-sample) setting; we thus consider them more as a motivation rather than a full justification of using multi-domain calibration in practice as we do. We look forward to expanding the scope of theoretical understanding of the conditions under which multi-domain calibration can provably guarantee out-of-domain generalization, including the finite-sample setting and the analysis of specific algorithms. We also expect new practical methods, building on our findings, will help push forward the real-world ability to generalize to unseen test domains.

# Acknowledgments

We wish to thank Ira Shavitt for his helpful comments and to Alexandre Ramé for pointing us to an error in the original manuscript. This research was partially supported by the Israel Science Foundation (grant No. 1950/19).

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
