# A   Proofs for Theoretical Claims

We begin by supplementing the definition of multiple domain calibration, extending it for the case of regression, then we provide proofs of the theorems in the paper.

## A.1   Definition of Calibration

Recall our definition of a calibrated classifier for binary tasks.

**Definition S1.** *Let $f : \mathcal{X} \to [0,1]$ and $P[X,Y]$ be a joint distribution over the features and label. Then $f(\mathbf{x})$ is calibrated w.r.t to $P$ if for all $\alpha \in [0,1]$ in the range of $f$:*

$$\mathbb{E}_P[Y \mid f(X) = \alpha] = \alpha.$$

*In the multiple environments setting, $f(\mathbf{x})$ is calibrated on $E_{train}$ if for all $e_i \in E_{train}$ and $\alpha$ in the range of $f$ restricted to $e_i$:*

$$\mathbb{E}[Y \mid f(X) = \alpha, E = e_i] = \alpha.$$

Let us prove the connection between multi-domain calibration and invariance, we repeat the statement of the lemma from the main paper for convenience.

**Lemma S1** (Lemma 1 in main paper)**.** *If a binary classifier $f$ is invariant w.r.t $E_{train}$ then there exists some $g : \mathbb{R} \to [0,1]$ such that $g \circ f$ is calibrated on all training environments and its mean squared error on each environment does not exceed that of $f$. On the other hand, if a classifier is calibrated on all training environments it is also invariant w.r.t $E_{train}$.*

*Proof.* Assume that the classifier is invariant w.r.t $E_{\text{train}}$, let $e_i \in E_{\text{train}}$ and note that:

$$\mathbb{E}[(Y - f(X))^2 \mid E = e_i] \geq \min_{g:\mathbb{R}\to\mathbb{R}} \mathbb{E}[(Y - g \circ f(X))^2 \mid E = e_i].$$

The solution to the RHS is to take $g(\hat{\alpha}) = \mathbb{E}[Y \mid f(X) = \hat{\alpha}, E = e_i]$ for all $\hat{\alpha} \in [0,1]$ and it results in a classifier $g \circ f$ that is calibrated w.r.t $e_i$. Due to invariance, for all $\hat{\alpha} \in \mathbb{R}$ the expectation $\mathbb{E}[Y \mid f(X) = \hat{\alpha}]$ is identical across all $e_i \in E_{\text{train}}$ where $\hat{\alpha}$ is in the range of $f$ restricted to $e_i$. Therefore there exists a single function $g$ that solves the RHS simultaneously over all environments. The resulting $g \circ f$ is indeed calibrated over all training domains and its mean squared error does not exceed that of $f$ (note that since the square loss is Bayes-consistent, this claim also holds for the classification error). The other part of the statement that a calibrated classifier on all $E_{\text{train}}$ is invariant follows easily from the definitions. $\qquad\square$

For regression tasks, one may consider a function that outputs a full CDF on $Y$ and define a calibrated classifier as one where all quantiles of the CDF match the true quantiles of $Y$ as the number of examples approached infinity. This leads to the definition in [25], and one may follow this to analyze more general cases than the scenario we will consider in this work.

Since in this section we consider Gaussian distributions and linear regressors, a definition based on the first two moments of the distribution (instead of all quantiles of a CDF) will suffice. Hence we will be working the following definition:

**Definition S2.** *Let $f : \mathcal{X} \to \mathbb{R}^2$ and $P[X,Y]$ a joint distribution over the features and label. Then $f(\mathbf{x})$ is calibrated w.r.t to $P$ if for all $(\alpha, \beta) \in \mathbb{R}^2$ in the range of $f$:*

$$\mathbb{E}[Y \mid f(X)_1 = \alpha] = \alpha, \; \mathbb{E}[Y^2 \mid f(X)_2 = \beta] = \beta.$$

*In the multiple environments setting, $f(\mathbf{x})$ is calibrated on $E_{train}$ if for all $e_i \in E_{train}$ and $(\alpha, \beta)$ in the range of $f$ restricted to $e_i$:*

$$\mathbb{E}[Y \mid f(X) = (\alpha, \beta), E = e_i] = \alpha, \; \mathbb{E}[Y^2 \mid f(X) = (\alpha, \beta), E = e_i] = \beta. \tag{4}$$

## A.2   Details about ECE, MMCE and Post-Processing Methods

To evaluate calibration and optimize our models towards multi-domain calibration, we use the Expected Calibration Error (ECE) and the Maximum Mean Calibration Error (MMCE) [26].

The ECE is a scalar summary of the calibration plot, used throughout the literature to assess how well calibrated is a given classifier. **Calibration plots** [7] are a visual representation of model calibration

in the case of binary labels. Each example $\mathbf{x}$ is placed into one of $B$ bins that partition the $[0, 1]$ interval, in which the output, or *confidence*, of the classifier $f(\mathbf{x})$ falls. For each bin $b$, the accuracy of $f$ on the bin's examples $acc(b)$ is calculated along with the average confidence $conf(b)$. These are plotted against each other to form a curve, where deviations from a diagonal represent miscalibration. **ECE score** summarizes the calibration curve by averaging the deviation between accuracy and confidence:

$$ECE = \sum_{b=1}^{B} \frac{n_b}{N} |acc(b) - conf(b)|. \tag{5}$$

$n_b$ is the number of examples in bin $b$, $N$ is the total number of examples. In all of our experiments we used $B = 10$ bins of equal size.

To handle the miscalibration that is often observed in models such as neural networks [12], the MMCE was proposed in [26] as a method to improve calibration at training time. Recalling the definition of this loss: We consider a dataset $D = \{\mathbf{x}_i, y_i\}_{i=1}^{m}$, a binary classifier parameterized by a vector $\theta$ which we denote $f_\theta :\to [0, 1]$. The confidence of $f_\theta$ on the $i$-th example is $f_{\theta;i} = \max\{f_\theta(x_i), 1 - f_\theta(x_i)\}$ and its correctness is $c_i = \mathbb{1}_{|y_i - f_{\theta;i}| < \frac{1}{2}}$. Then we fix a kernel $k : \mathbb{R} \times \mathbb{R} \to \mathbb{R}$, associated with a feature map $\phi : [0, 1] \to \mathcal{H}$, and MMCE over the dataset $D$ is given by:

$$r_{\text{MMCE}}^{D}(f_\theta) = \frac{1}{m^2} \sum_{i,j \in D} (c_i - f_{\theta;i})(c_j - f_{\theta;j}) k(f_{\theta;i}, f_{\theta;j}). \tag{6}$$

In our experiments we use an RBF kernel $k(r, r') = \exp(-\gamma(r - r')^2)$ with $\gamma = 2.5$. Equation (6) is the finite sample approximation of the following:

$$MMCE(f_\theta; P[X, Y]) = \|\mathbb{E}_{(\mathbf{x},y) \sim P}[(c - f_\theta(\mathbf{x}))\phi(f_\theta(\mathbf{x}))]\|_{\mathcal{H}}. \tag{7}$$

Here $c$ is the correctness of $f_\theta$ on $(\mathbf{x}, y)$ as defined for Equation (6). Attractive properties of the MMCE include it being a proper scoring rule:

**Theorem** (Adapted from Thm. 1 in [26]). *Let $P[X, Y]$ be a probability measure defined on the space $(\mathcal{X} \times \{0, 1\})$ such that the conditionals on the pushforward measure $P[r, c] = f_\theta \sharp P$,[6] $P(r \mid c = 1)$ over $([0, 1] \times \{0, 1\})$, $P(r \mid c = 0)$ are Borel probability measures, and let $k$ be a universal kernel. The MMCE in Equation (7) is $0$ if and only if $f_\theta$ is calibrated w.r.t $P$.*

Corollary 1 in the paper follows by considering $\sum_{e \in E_{\text{train}}} MMCE(f_\theta; P[X, Y \mid E = e])$ and applying the theorem to each summand. For more details on the MMCE, its derivation as an integral probability measure analogue of the ECE and its properties, we refer the reader to [26].

Another popular metric for calibration in binary classification problems is the Brier score, which is simply the squared error between the predicted probability and the outcome [5]:

$$BS(f) = \frac{1}{m} \sum_{i=1}^{m} (f(\mathbf{x}_i) - y_i)^2.$$

The Isotonic Regression [30] post-processing methods that we use in the paper minimize the Brier score using a monotonic post-processing function. Hence we consider a classifier $f$ and a dataset $\{\mathbf{x}_i, y_i\}_{i=1}^{m}$. Denote the prediction of $f$ on $\mathbf{x}_i$ by $f_i$, then isotonic regression solves:

$$\min_{z : f_i \leq f_j \Rightarrow z(f_i) \leq z(f_j)} \frac{1}{m} \sum_{i=1}^{m} (z(f_i) - y_i)^2.$$

A motivation for using this as a post-processing calibration method is the decomposition of the Brier score to a refinement and calibration score. We may denote the set of prediction values that are obtained by $f$ across the dataset by $F = \{f_i \mid i \in [m]\}$. For each such value $\tilde{f} \in F$ then denote $N_{\tilde{f}} = |\{i \mid f_i = \tilde{f}\}|$ as the number of points for which we obtain this prediction and $y_{\tilde{f}} = \frac{1}{N_{\tilde{f}}} \sum_{i:f_i=\tilde{f}} y_i$ the average outcome over them:

$$BS(f) = CAL(f) + REF(f) = \frac{1}{m} \sum_{\tilde{f} \in F} N_{\tilde{f}}(\tilde{f} - y_{\tilde{f}})^2 + \frac{1}{m} \sum_{\tilde{f} \in F} N_{\tilde{f}}(y_{\tilde{f}}(1 - y_{\tilde{f}}))$$

---

[6]we note the abuse of notation here, as $f_\theta \sharp P$ is used to denote the measure that we get by applying $f_\theta$ to $X$ to obtain $r$ and $c$ is obtained by calculating its correctness w.r.t to $Y$.

The calibration score measures how far is the average prediction value from the average outcome, while refinement gives a measure of their sharpness (i.e. it raises the score of uncertain prediction). Due to the monotonicity constraint of isotonic Regression, it is usually thought of as not changing the $REF(f)$ too much, which means it minimizes the Brier score mainly by reducing $CAL(f)$. In the multi-domain cases we are interested in, note that this vanilla isotonic regression does not take domains into account. In our experiments we use it simple by pooling the dataset on all environments and performing post-processing calibration on this dataset using isotonic regression. This procedure could output a classifier that is perfectly calibrated for the entire dataset, but not on single environments.

To give a simple variant that does post-processing while taking environments into account, we proposed a Robust Isotonic Regression method. The method minimizes the Brier score on the worst-case environment, thus aiming to bound the worst miscalibration on each environment. While in practice it will usually not provide perfect calibration on each environment, the method trades off the error between environments so it is better geared towards simultaneous calibration of the classifier on all domains. Formally we solve:

$$z^* = \underset{z: f_i \leq f_j \Rightarrow z(f_i) \leq z(f_j)}{\arg\min} \max_{e \in E_{\text{train}}} \frac{1}{N_e} \sum_{i=1}^{N_e} (z(f_{e,i}) - y_i)^2. \tag{8}$$

Where $N_e$ are the number of data points in environment $e \in E_{\text{train}}$ and $f_{e,i}$ is the output of $f$ on point $i$ in the environment.

### A.3 Causal Graphical Models

In order to answer queries about unseen distributions based on data from different, observed distributions, one must make certain assumptions about the data generating processes and the relationships between the observed and unobserved distributions. One way of articulating such models of the world is by using causal graphs. In a causal graph, edges from a variable $X$ to a variable $Y$ mean that changing the value of $X$ *may* change the distribution of $Y$. Causal graphs entail all statistical dependencies between variables, and we can read off such independence statements using the d-separation criterion [32]. We refer to background material to discuss how to identify and estimate causal effects with these causal graphical models in hand [33].

In the main paper, Figure 1 illustrates our assumed causal graph for a general problem of distribution shift, and Figure 2 illustrates the assumed causal graph for causal and anti-causal simplified examples described in equations 2 and 3, respectively. For instance according to d-separation, in distributions described by Figure 1 it holds that $Y \perp\!\!\!\perp E \mid X_{\text{causal}}, X_{\text{ac-non-spurious}}$ and that in general $Y \not\perp\!\!\!\perp E \mid X_{\text{ac-spurious}}$. Furthermore, if we introduce a node $\Phi(\mathcal{X})$ whose parents do not include $X_{\text{ac-spurious}}$, then $Y \perp\!\!\!\perp E \mid \Phi(X)$ (and conversely, if $X_{\text{ac-spurious}} is a parent then the independence does not hold in general), which motivates the definition of a representation tha$

Equipped with the definitions and background given in the previous sections, we now turn to the proofs of the theorems in the paper.

### A.4 Classification with Invariant Features

We first consider the classification task from the main paper, where the data generating process is described in Figure S1. Recall that we are considering linear classifiers of the form $f(\mathbf{x}; \mathbf{w}, b) = \sigma(\mathbf{w}^\top \mathbf{x} + b)$. Our environments here are defined by the parameters of the multivariate Gaussian distributions that generate the spurious features $\{\mu_i, \Sigma_i\}_{i=1}^k$. As a first step we will derive the algebraic form of the constraints that calibration imposes on $\mathbf{w}$ and the parameters defining the environments. For convenience, we modify the notation from the main paper and consider a binary label where $\mathcal{Y} = \{-1, 1\}$ instead of $\mathcal{Y} = \{0, 1\}$.

**Lemma S2.** *Assume we have $k$ environments with means and covariance matrices for environmental features $\mu_i \in \mathbb{R}^{d_e}, \Sigma_i \in \mathbb{S}_{++}^{d_e}, i \in [k]$ and a common covariance matrix $\Sigma_{ns} \in \mathbb{S}_{++}^{d_{ns}}$ for invariant features, where data is generated according to:*

$$y = \begin{cases} 1 & \text{w.p } \eta \\ -1 & \text{otherwise} \end{cases}, \qquad \begin{aligned} \mathbf{x}_{ns} \mid Y = y &\sim \mathcal{N}(y\mu_{ns}, \Sigma_{ns}), \\ \mathbf{x}_{sp} \mid Y = y &\sim \mathcal{N}(y\mu_i, \Sigma_i), \end{aligned}$$

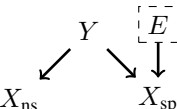

$$Y \quad E$$
$$X_{\mathrm{ns}} \quad X_{\mathrm{sp}}$$

Figure S1: Diagram for data generating process in the invariant features scenario.

*and $\mathbf{x}_{ns}, \mathbf{x}_{sp}$ are drawn independently. Let $\sigma : \mathbb{R} \to (0, 1)$ be an invertible function and define the classifier:*

$$f(\mathbf{x}; \mathbf{w}, b) = \sigma(\mathbf{w}^\top \mathbf{x} - b).$$

*Decompose the weights $\mathbf{w} = [\mathbf{w}_{ns}, \mathbf{w}_{sp}]$ to the coefficients of the invariant and spurious features accordingly. Then if the classifier is calibrated on all environments, it holds that either $\mathbf{w} = \mathbf{0}$ or there exists $t \neq 0$ such that:*

$$\frac{\mathbf{w}_{ns}^\top \mu_{ns} + \mathbf{w}_{sp}^\top \mu_i}{\mathbf{w}_{ns}^\top \Sigma_{ns} \mathbf{w}_{ns} + \mathbf{w}_{sp}^\top \Sigma_i \mathbf{w}_{sp}} = t \quad \forall i \in [k]. \tag{9}$$

*Proof.* Let $i \in [k]$, the joint distribution of features in the environment is Gaussian with mean $\hat{\mu}_i = [\mu_{\mathrm{ns}}, \mu_i]$, covariance $\hat{\Sigma}_i = \begin{bmatrix} \Sigma_{\mathrm{ns}} & 0 \\ 0 & \Sigma_i \end{bmatrix}$. Hence the output of the affine function corresponding to the classifier is a random variable with probability density function:

$$P[\sigma^{-1}(f(X)) = \alpha \mid Y = y, E = e_i] = (2\pi \mathbf{w}^\top \hat{\Sigma}_i \mathbf{w})^{-\frac{1}{2}} \exp\left( \frac{\left(\alpha - y\mathbf{w}^\top \hat{\mu}_i + b\right)^2}{2\mathbf{w}^\top \hat{\Sigma}_i \mathbf{w}} \right).$$

Hence the conditional probability of $Y$ is given by:

$$P[Y = 1 \mid \sigma^{-1}(f(X)) = \alpha, E = e_i] = \frac{\eta \exp\left( \frac{(\alpha - \mathbf{w}^\top \hat{\mu}_i + b)^2}{2\mathbf{w}^\top \hat{\Sigma}_i \mathbf{w}} \right)}{\eta \exp\left( \frac{(\alpha - \mathbf{w}^\top \hat{\mu}_i + b)^2}{2\mathbf{w}^\top \hat{\Sigma}_i \mathbf{w}} \right) + (1 - \eta) \exp\left( \frac{(\alpha + \mathbf{w}^\top \hat{\mu}_i + b)^2}{2\mathbf{w}^\top \hat{\Sigma}_i \mathbf{w}} \right)}.$$

Note that unless $\mathbf{w} = \mathbf{0}$ (which results in a calibrated classifier that satisfies Equation (9)), the variance of $\sigma^{-1}(f(X))$ is strictly positive since $\hat{\Sigma}_i \succ 0$, so above conditional probabilities are well-defined. Now it is easy to see that if the classifier is calibrated across environments, we need to have equality in the log-odds ratio for each $i, j$ and all $\alpha \in \mathbb{R}$:

$$\frac{\left(\alpha - \mathbf{w}^\top \hat{\mu}_i + b\right)^2}{2\mathbf{w}^\top \hat{\Sigma}_i \mathbf{w}} - \frac{\left(\alpha + \mathbf{w}^\top \hat{\mu}_i + b\right)^2}{2\mathbf{w}^\top \hat{\Sigma}_i \mathbf{w}} = \frac{\left(\alpha - \mathbf{w}^\top \hat{\mu}_j + b\right)^2}{2\mathbf{w}^\top \hat{\Sigma}_j \mathbf{w}} - \frac{\left(\alpha + \mathbf{w}^\top \hat{\mu}_j + b\right)^2}{2\mathbf{w}^\top \hat{\Sigma}_j \mathbf{w}} \quad \forall \alpha \in \mathbb{R}.$$

After dropping all the terms that cancel out in the subtractions we arrive at:

$$\frac{\mathbf{w}^\top \hat{\mu}_i}{\mathbf{w}^\top \hat{\Sigma}_i \mathbf{w}} = \frac{\mathbf{w}^\top \hat{\mu}_j}{\mathbf{w}^\top \hat{\Sigma}_j \mathbf{w}}.$$

This may also be written as a system of equations with an additional scalar variable $t \in \mathbb{R}$:

$$\frac{\mathbf{w}^\top \hat{\mu}_i}{\mathbf{w}^\top \hat{\Sigma}_i \mathbf{w}} = t \quad \forall i \in [k].$$

Now because we assumed $\Sigma_i \succ 0$ for all environments, for any solution to the above system with $t = 0$, we must have:

$$\mathbf{w}^\top \hat{\mu}_i = 0 \quad \forall i \in [k].$$

Furthermore we will have for any $\alpha \in \mathbb{R}$:

$$P[Y = 1 \mid \sigma^{-1}(f(X)) = \alpha, E = e_i] = \eta.$$

Since we assume $f$ is calibrated and the right hand side needs to equal $\alpha$, this is only possible if $f(\mathbf{x}; \mathbf{w}, b)$ is a constant function. Again, because $\Sigma_i \succ 0$, this is only possible if $\mathbf{w} = \mathbf{0}$. Hence we conclude with our desired result, as can be seen by decomposing $\mathbf{w}$ to the parts corresponding to invariant and spurious features. $\quad\square$

We now give a result for the special case where the covariance matrices of the spurious features satisfy $\Sigma_i = \sigma_i^2 \mathbf{I}$, considered in [37]. The nice correspondence here is that we will see that calibration demands one more environment than IRM to discard all spurious features. This matches the intuition that each environment reduces a degree of freedom from the set of invariant classifiers, while risk minimization reduces one more degree of freedom.

**Lemma S3.** *Assume we have $k \geq d_{sp} + 2$ environments and define $M\left(\{\mu_i, \sigma_i\}_{i=1}^k\right) \in \mathbb{R}^{k \times d_e + 2}$:*

$$
M(\{\mu_i, \sigma_i\}_{i=1}^k) = \begin{bmatrix} \mu_1^\top & \sigma_1^2 & 1 \\ & \vdots & \\ \mu_k^\top & \sigma_k^2 & 1 \end{bmatrix}.
$$

*If the matrix has full rank, then for any invariant predictor the linear coefficients on spurious features are zero.*

*Proof.* According to Lemma S2, writing down the conditional probability $P[Y \mid \sigma^{-1}(f(\mathbf{x})), E = e]$ and demanding calibration results in the constraint that either $\mathbf{w} = \mathbf{0}$, and then the linear coefficients on spurious features are indeed 0; or that for some $t \neq 0$:

$$
\frac{\mathbf{w}_{\mathrm{ns}}^\top \mu_{\mathrm{ns}} + \mathbf{w}_{\mathrm{sp}}^\top \mu_i}{\mathbf{w}_{\mathrm{ns}}^\top \Sigma_{\mathrm{ns}} \mathbf{w}_{\mathrm{ns}} + \sigma_i^2 \|\mathbf{w}_{\mathrm{sp}}\|_2^2} = t \quad \forall i \in [k].
$$

Without loss of generality we can phrase these constraints as:

$$
\frac{\mathbf{w}_{\mathrm{ns}}^\top \mu_{\mathrm{ns}} + \mathbf{w}_{\mathrm{sp}}^\top \mu_i}{\mathbf{w}_{\mathrm{ns}}^\top \Sigma_{\mathrm{ns}} \mathbf{w}_{\mathrm{ns}} + \sigma_i^2 \|\mathbf{w}_{\mathrm{sp}}\|_2^2} = 1 \quad \forall i \in [k].
$$

This is true since if $\mathbf{w}$ is a solution to this system of equations where the right hand side is some $t \in \mathbb{R}$ then $t\mathbf{w}$ is a solution to the system where $t$ is replaced by 1. Rewrite the constraints again to isolate the parts depending on $\mathbf{w}_{\mathrm{sp}}$:

$$
\sigma_i^2 \|\mathbf{w}_{\mathrm{sp}}\|_2^2 - \mu_i^\top \mathbf{w}_{\mathrm{sp}} = \mathbf{w}_{\mathrm{ns}}^\top \Sigma_{\mathrm{ns}} \mathbf{w}_{\mathrm{ns}} - \mathbf{w}_{\mathrm{ns}}^\top \mu_{\mathrm{ns}} \quad \forall i \in [k].
$$

To find whether this system has a solution where $\mathbf{w}_{\mathrm{sp}}$ is non-zero we can replace the right hand side with a scalar variable $t \in \mathbb{R}$, and ask whether the following system has a non-zero solution:

$$
\sigma_i^2 \|\mathbf{w}_{\mathrm{sp}}\|_2^2 - \mu_i^\top \mathbf{w}_{\mathrm{sp}} = t \quad \forall i \in [k].
$$

For the above equations to have a non-zero solution, the following linear system must also have such a solution:

$$
M(\{\mu_i, \sigma_i\}_{i=1}^k)\mathbf{x} = \mathbf{0}.
$$

But from our non-degeneracy condition, such a solution does not exist. $\qquad\square$

Next we generalize the above to prove the result from the main paper, namely when the matrices $\{\Sigma_i\}_{i=1}^k$ are not diagonal. For this purpose we introduce a definition of general position for environments, similar to the one given in [1].

**Definition S3.** *Given $k > 2d_{sp}$ environments with mean parameters $\{\Sigma_i, \mu_i\}_{i=1}^k$, we say they are in general position if for all non-zero $\mathbf{x} \in \mathbb{R}_{sp}^d$:*

$$
\dim\left(\mathrm{span}\left\{\begin{bmatrix} \Sigma_i \mathbf{x} + \mu_i \\ 1 \end{bmatrix}\right\}_{i \in [k]}\right) = d_e + 1.
$$

Equipped with this notion of general position, we now need to show that if it holds then the only predictors that satisfy the conditions of Lemma S2 are those with $\mathbf{w}_{\mathrm{sp}} = \mathbf{0}$. Another claim we will need to prove is that the subset of environments which do not lie in general position have measure zero in the set of all possible environment settings. Hence generic environments are expected to lie in general position. This argument will follow the lines of the one given in [1], adapted to our case with the fixed coordinate 1 added in the above definition.

**Theorem 1.** *Under the setting of Lemma S2, if the environments lie in general position then all classifiers that are calibrated across environments satisfy $\mathbf{w}_{sp} = \mathbf{0}$.*

*Proof.* According to Lemma S2, if the predictor is calibrated then Equation (9) must hold. Following the same arguments laid out in the proof at the main paper, we get that $\mathbf{w}_{sp}$ needs to be a solution for the following system of equations:

$$\mathbf{w}_{sp}^\top \Sigma_i \mathbf{w}_{sp} - \mu_i^\top \mathbf{w}_{sp} - t = 0 \quad \forall i \in [k]. \tag{10}$$

Now, let $\mathbf{w}_{sp} \in \mathbb{R}^{d_{\mathrm{sp}}}$ be a non-zero vector and let us define the $k \times d_e + 1$ matrix:

$$M(\{\mu_i, \Sigma_i\}_{i=1}^k, \mathbf{w}_{sp}) = \begin{bmatrix} \mathbf{w}_{sp}^\top \Sigma_1 - \mu_1^\top & 1 \\ \vdots & \\ \mathbf{w}_{sp}^\top \Sigma_k - \mu_k^\top & 1 \end{bmatrix}$$

If the environments are in general position, the above matrix has full rank for any non-zero $\mathbf{w}_{sp}$. Similarly to the proof of Lemma S3, if Equation (10) has a non-zero solution then the following system must also have a solution:

$$M(\{\mu_i, \Sigma_i\}_{i=1}^k, \mathbf{w}_{sp})\mathbf{x} = \mathbf{0}.$$

Which is of course impossible due to $M(\{\mu_i, \Sigma_i\}_{i=1}^k, \mathbf{w}_{sp})$ having full rank. $\square$

We conclude with the statement about the measure of sets of environments which do not lie in general position, this will follow the lines of [1].

**Lemma S4.** *Let $k > 2d_{sp}$ and $\{\mu_i\}_{i=1}^k$ be arbitrary fixed vectors, then the set of matrices $\{\Sigma_i\}_{i=1}^k \in (\mathbb{S}_{++}^{d_{sp}})^k$ for which $\{\Sigma_i, \mu_i\}_{i=1}^k$ do not lie in general position has measure zero within the set $(\mathbb{S}_{++}^{d_{sp}})^k$.*

*Proof.* We assume $k > 2d_{\mathrm{sp}}$ and denote by $LR(k, d_{\mathrm{sp}}, r)$ the matrices of dimensions $k \times d_{\mathrm{sp}}$ and rank $r$. Also for any $d$ denote by $\mathbf{1}_d$ the vector in $\mathbb{R}^d$ where all entries equal 1. Define $\mathbf{M}_*^1(k, d_{\mathrm{sp}})$ as the set of $k \times d_{\mathrm{sp}}$ matrices of full column-rank whose columns span the vector of ones $\mathbf{1}_k$:

$$\mathbf{M}_*^1(k, d_{\mathrm{sp}}) = \{A \in LR(k, d_{\mathrm{sp}}, d_{\mathrm{sp}}) \mid \mathbf{1}_k \in \mathrm{colsp}(A)\}.$$

Let $\{\Sigma_i\}_{i=1}^k \in (\mathbb{S}_{++}^{d_{\mathrm{sp}}})^k$ and define $\mathbf{W} \subseteq \mathbb{R}^{k \times d_{sp}}$ as the image of the mapping $G : \mathbb{R}^{d_{sp}} \setminus \{0\} \to \mathbb{R}^{k \times d_{sp}}$:

$$(G(\mathbf{x}))_{i,l} = (\Sigma_i \mathbf{x} - \mu_i)_l$$

By the definition of general position given in the paper, the environments defined by $\{\Sigma_i, \mu_i\}_{i=1}^k$ lie in general position if $\mathbf{W}$ does not intersect $LR(k, d_{\mathrm{sp}}, r)$ for all $r < d_{\mathrm{sp}}$ and $\mathbf{M}_*^1(k, d_{\mathrm{sp}})$. We would like to show that this happens for all but a measure zero of $\left(\mathbb{S}_{++}^{d_{sp}}\right)^k$.

Due to the exact same arguments in Thoerem 10 of [1], we have that $\mathbf{W}$ is transversal to any submanifold of $\mathbb{R}^{k \times d_{\mathrm{sp}}}$ and also does not intersect $LR(k, d_{\mathrm{sp}}, r)$ where $r < d_{\mathrm{sp}}$, for all $\{\Sigma_i\}_{i=1}^k$ but a measure zero of $\left(\mathbb{S}_{++}^{d_{sp}}\right)^k$.

It is left to show that it also does not intersect $\mathbf{M}_*^1(k, d_{\mathrm{sp}})$ for all but a measure zero of $\left(\mathbb{S}_{++}^{d_{sp}}\right)^k$. Because $\mathbf{M}_*^1(k, d_{\mathrm{sp}})$ is a submanifold of $\mathbb{R}^{k \times d_{\mathrm{sp}}}$, it intersects transversally with $\mathbf{W}$ for generic $\{\Sigma_i\}_{i=1}^k$. Then by transversality they cannot intersect if $\dim(\mathbf{W}) + \dim(\mathbf{M}_*^1(k, d_{\mathrm{sp}})) - \dim(\mathbb{R}^{k \times d_{\mathrm{sp}}}) < 0$. We will claim that $\dim(\mathbf{M}_*^1(k, d_{\mathrm{sp}})) = k(d_{\mathrm{sp}} - 1) + d_{\mathrm{sp}}$ and then since $k > 2d_{\mathrm{sp}}$ we may obtain:

$$\dim(\mathbf{W}) + \dim(\mathbf{M}_*^1(k, d_{\mathrm{sp}})) - \dim(\mathbb{R}^{k \times d_{\mathrm{sp}}}) \leq d_{\mathrm{sp}} + k(d_{\mathrm{sp}} - 1) + d_{\mathrm{sp}} - kd_{\mathrm{sp}}$$
$$= 2d_{\mathrm{sp}} - k$$
$$< 0.$$

The negativity of the dimension implies that if $\mathbf{W}$ and $\mathbf{M}_*^1(k, d_{\mathrm{sp}})$ are transversal then they do not intersect, and we may conclude our desired result that the environments lie in general position for all but a measure zero of $\left(\mathbb{S}_{++}^{d_{\mathrm{sp}}}\right)^k$.

To show that $\dim(\mathbf{M}_*^1(k, d_{\mathrm{sp}})) = k(d_{\mathrm{sp}} - 1) + d_{\mathrm{sp}}$, consider a matrix $A \in \mathbf{M}_*^1(k, d_{\mathrm{sp}})$. Since it has full rank, it has a $d_{\mathrm{sp}} \times d_{\mathrm{sp}}$ minor that is invertible. Assume this minor is just the first $d_{\mathrm{sp}}$ rows of $A$, otherwise there is a linear isomorphism that transforms it into such a matrix and the arguments that follow still apply (see [27], Example 5.30; our proof follows a similar line of reasoning). Now write $A$ as a block matrix using $B \in \mathbb{R}^{d_{\mathrm{sp}} \times d_{\mathrm{sp}}}, C \in \mathbb{R}^{(k - d_{\mathrm{sp}}) \times d_{\mathrm{sp}}}$:

$$A = \begin{bmatrix} B \\ C \end{bmatrix}.$$

Denoting by $\mathbf{U}$ the set of $k \times d_{\mathrm{sp}}$ matrices whose first $d_{\mathrm{sp}}$ rows are invertible, we consider the mapping $F : \mathbf{U} \to \mathbb{R}^{k - d_{\mathrm{sp}}}$:

$$F(A) = \mathbf{1}_{k - d_{\mathrm{sp}}} - C B^{-1} \mathbf{1}_{d_{\mathrm{sp}}}.$$

Clearly $F^{-1}(\mathbf{0}) = \mathbf{M}_*^1(k, d_{\mathrm{sp}})$ and $F$ is smooth. We will show that it is a submersion by observing that its differential $DF(U)$ is surjective for each $U \in \mathbf{U}$. To this end, for a given $U = \begin{bmatrix} B \\ C \end{bmatrix}$ and any $X \in \mathbb{R}^{(k - d_{\mathrm{sp}}) \times d_{\mathrm{sp}}}$ define a curve $\gamma : (-\epsilon, \epsilon) \to \mathbf{U}$ by:

$$\gamma(t) = \begin{bmatrix} B \\ C + \gamma X \end{bmatrix}.$$

We have that:

$$(F \circ \gamma)'(t) = \frac{d}{dt}\big|_{t=0} (\mathbf{1}_{k - d_{\mathrm{sp}}} - (C + tX) B^{-1} \mathbf{1}_{d_{\mathrm{sp}}}) = X B^{-1} \mathbf{1}_{d_{\mathrm{sp}}}.$$

Since $B^{-1} \mathbf{1}_{d_{\mathrm{sp}}}$ is not the zero vector, and $X \in \mathbb{R}^{(k - d_{\mathrm{sp}}) \times d_{\mathrm{sp}}}$ where $k - d_{\mathrm{sp}} > d_{\mathrm{sp}}$, then it is clear that the above mapping is surjective. Note that the derivatives along the curve are just a subset of the range of $DF(U)$, hence $DF(U)$ is also surjective at each point $U \in \mathbf{U}$. It follows from the submersion theorem that $\dim(\mathbf{M}_*^1(k, d_{\mathrm{sp}})) = k d_{\mathrm{sp}} - (k - d_{\mathrm{sp}}) = k(d_{\mathrm{sp}} - 1) + d_{\mathrm{sp}}$ as desired for our result to hold. $\qquad \square$

## A.5 Regression Under Covariate Shift and Spurious Features

We now move on to the second scenario presented in the paper where the mechanism $P(Y \mid X)$ is invariant and the diagram depicting the data generating process is given in Figure S2. Here for each environment $i \in [k]$ we will have:

$$\begin{aligned} X_c &\sim \mathcal{N}(\mu_i^c, \Sigma_i^c) \qquad\qquad\qquad\qquad\qquad\qquad\qquad (11) \\ Y &= \mathbf{w}_c^{*\top} \mathbf{x}_c + \xi, \; \xi \sim \mathcal{N}(0, \sigma_y^2) \\ X_{sp} &= y \mu_i + \eta, \; \eta \sim \mathcal{N}(\mathbf{0}, \Sigma_i). \end{aligned}$$

We consider a regressor $f : \mathcal{X} \to \mathbb{R}^2$, where the estimate of the mean is linear, i.e. $[f(\mathbf{x}; \mathbf{w})]_1 = \mathbf{w}^\top \mathbf{x}$, and the estimate of the variance is constant $[f(\mathbf{x}; \mathbf{w})]_2 = c$.[7] We decompose the weights $\mathbf{w}$ into their parts corresponding to causal and spurious features $[\mathbf{w}_c, \mathbf{w}_{sp}]$. Then our result regarding calibration and generalization to $\mathcal{E}$ is given below.

**Theorem 2.** *Denote the dimensions of $X_c, X_{sp}$ by $d_c, d_{sp}$ accordingly. Assume we have $k$ environments with parameters $\{\mu_i^c, \mu_i, \Sigma_i^c, \Sigma_i\}_{i=1}^k$. For any matrix $A$ denote its $i$-th row by $A^i$, and define the matrices $M(\{\mu_i^c, \mu_i\}_{i=1}^k) \in \mathbb{R}^{k \times d_c + d_{sp} + 1}$ and $M_2(\{\mu_i^c, \Sigma_i^c\}_{i=1}^k, \sigma_y^2, \mathbf{w}_c^*) \in \mathbb{R}^{k \times d_c + 2}$ whose rows*

---

[7]Limiting the variance estimate to a constant does not make a difference for the purpose of our proof. The proof does not rely on the correctness of the variance estimate as imposed by Equation (4), but only on the variances being equal across environments when conditioned on $f(\mathbf{x})$. In other words it relies on the correctness of the mean estimate, and the distribution of $Y$ conditioned on $f(X)$ being the same across environments.

$$X_c \longrightarrow Y \longrightarrow X_{\text{ac-sp}}$$

Figure S2: Diagram for data generating process in the covariate shift scenario.

*are given by:*

$$M(\{\mu_i^c, \mu_i\}_{i=1}^k) = \begin{bmatrix} \mu_i^{c\top} & \left(\mathbf{w}_c^{*\top}\mu_1^c\right)\mu_1^\top & 1 \\ & \vdots & \\ \mu_k^{c\top} & \left(\mathbf{w}_c^{*\top}\mu_k^c\right)\mu_k^\top & 1 \end{bmatrix},$$

$$M_2(\{\mu_i^c, \Sigma_i^c\}_{i=1}^k, \sigma_y^2, \mathbf{w}_c^*) = \begin{bmatrix} \mathbf{w}_c^{*\top}\Sigma_1^c + \left(\frac{\mathbf{w}_c^{*\top}\Sigma_1^c\mathbf{w}_c^* + \sigma_y^2}{\mathbf{w}_c^{*\top}\mu_1^c}\right)\mu_1^{c\top} & \frac{\mathbf{w}_c^{*\top}\Sigma_1^c\mathbf{w}_c^*}{\mathbf{w}_c^{*\top}\mu_1^c} & 1 \\ & \vdots & \\ \mathbf{w}_c^{*\top}\Sigma_k^c + \left(\frac{\mathbf{w}_c^{*\top}\Sigma_k^c\mathbf{w}_c^* + \sigma_y^2}{\mathbf{w}_c^{*\top}\mu_k^c}\right)\mu_k^{c\top} & \frac{\mathbf{w}_c^{*\top}\Sigma_k^c\mathbf{w}_c^*}{\mathbf{w}_c^{*\top}\mu_k^c} & 1 \end{bmatrix}.$$

*Let $f(\mathbf{x}; \mathbf{w})$ be a calibrated regressor, assume $\mathbf{w}_c^{*\top}\mu_i^c \neq 0$ for all $i \in [k]$ and that there exists $i, j \in [k]$ such that $\mathbb{E}[Y \mid E = e_i] \neq \mathbb{E}[Y \mid E = e_j]$. Furthermore assume that one of the following conditions hold:*

- *$k > \max\{d_c + 2, d_{sp}\}$, $M_2(\{\mu_i^c, \Sigma_i^c\}_{i=1}^k, \sigma_y^2, \mathbf{w}_c^*)$ has full rank and the means of spurious features $\{\mu_i\}_{i=1}^k$ span $\mathbb{R}^{d_{sp}}$.*

- *$k > d_c + d_{sp} + 1$ and $M(\{\mu_i^c, \mu_i\}_{i=1}^k)$ has full rank.*

*then the weights of $f$ must be $\mathbf{w} = [\mathbf{w}_c^*, \mathbf{0}]$.*

It is rather clear that rank-deficiency of $M_2$ would impose some highly non-trivial conditions on the relationships between $\mu_i^c, \mathbf{w}_c^{*\top}\Sigma_i^c$ and the conditions given above are satisfied for all settings of environments other than a measure zero under any absolutely continuous measure on the parameters $\mathbf{w}_c^*, \{\mu_i^c, \Sigma_i^c\}_{i=1}^k$. The proof proceeds by writing the conditional distribution of $Y$ on $f(X)$, and showing that the conditions in the theorem are the direct result of the calibration constraints.

*Proof.* Since $X_c, X_{sp}, Y$ are jointly Gaussian, we can write their distribution at environment $i \in [k]$ as:

$$\begin{bmatrix} X_c \\ X_{sp} \\ Y \end{bmatrix} \sim \mathcal{N}\left( \begin{bmatrix} \mu_i^c \\ (\mathbf{w}_c^{*\top}\mu_i^c)\mu_i \\ \mathbf{w}_c^{*\top}\mu_i^c \end{bmatrix}, \right.$$
$$\left. \begin{bmatrix} \Sigma_i^c & \Sigma_i^c\mathbf{w}_c^*\mu_i^\top & \Sigma_i^c\mathbf{w}_c^* \\ \mu_i\mathbf{w}_c^{*\top}\Sigma_i^c & \left(\mathbf{w}_c^{*\top}\Sigma_i^c\mathbf{w}_c^* + \sigma_y^2\right)\mu_i\mu_i^\top + \Sigma_i & \left(\mathbf{w}_c^{*\top}\Sigma_i^c\mathbf{w}_c^* + \sigma_y^2\right)\mu_i \\ \mathbf{w}_c^{*\top}\Sigma_i^c & (\mathbf{w}_c^{*\top}\Sigma_i^c\mathbf{w}_c^* + \sigma_y^2)\mu_i^\top & \mathbf{w}_c^{*\top}\Sigma_i^c\mathbf{w}_c^* + \sigma_y^2 \end{bmatrix} \right).$$

The predictions $\mathbf{w}^\top X$ are then also normally distributed, and jointly with $Y$ this can be written as:

$$\begin{bmatrix} \mathbf{w}^\top X \\ Y \end{bmatrix} \sim \mathcal{N}\left( \begin{bmatrix} \mathbf{w}_c^\top\mu_i^c + (\mathbf{w}_{sp}^\top\mu_i)(\mathbf{w}_c^{*\top}\mu_i^c) \\ \mathbf{w}_c^{*\top}\mu_i^c \end{bmatrix}, \begin{bmatrix} \sigma_{f,i}^2 & \sigma_{f,y,i} \\ \sigma_{f,y,i} & \sigma_{y,i}^2 \end{bmatrix} \right),$$

where we defined the items of the covariance matrix:

$$\sigma_{f,i}^2 = \mathbf{w}_c^\top\Sigma_i^c\mathbf{w}_c + 2(\mathbf{w}_c^\top\Sigma_i^c\mathbf{w}_c^*)(\mu_i^\top\mathbf{w}_{sp}) + \mathbf{w}_{sp}^\top\left(\mu_i\mu_i^\top(\mathbf{w}_c^{*\top}\Sigma_i^c\mathbf{w}_c^* + \sigma_y^2) + \Sigma_i\right)\mathbf{w}_{sp},$$

$$\sigma_{f,y,i} = \mathbf{w}_c^{*\top}\Sigma_i^c\mathbf{w}_c + (\mathbf{w}_c^{*\top}\Sigma_i^c\mathbf{w}_c^* + \sigma_y^2)\mu_i^\top\mathbf{w}_{sp},$$

$$\sigma_{y,i}^2 = \mathbf{w}_c^{*\top}\Sigma_i^c\mathbf{w}_c^* + \sigma_y^2.$$

Now we can write the mean of the conditional distribution of $Y$ on $f(X)_1 = \alpha$ as:

$$\mathbb{E}\left[Y \mid f(X)_1 = \alpha, E = e_i\right] = \mathbf{w}_c^{*\top}\mu_i^c + \frac{\sigma_{f,y,i}}{\sigma_{f,i}^2}(\alpha - \mathbf{w}_c^\top\mu_i^c - (\mathbf{w}_{sp}^\top\mu_i)(\mathbf{w}_c^{*\top}\mu_i^c)).$$

For each environment $i \in [k]$, the above is a linear function of $\alpha$. Demanding $f(X)$ to be calibrated on all environments then imposes both the slopes and intercepts to be equal across environments. Writing this for the slope, we obtain that there must exist $t \in \mathbb{R}$ such that:

$$\frac{\sigma_{f,y,i}}{\sigma_{f,i}^2} = t \quad \forall i \in [k]. \tag{12}$$

We note that $t \neq 0$ since if it is zero then we have that $\mathbb{E}[Y \mid f(X)_1 = \alpha, E = i]$ does not depend on $\alpha$, where calibration demands that it equals $\alpha$. This can only happen if $\mathbf{w}_c = \mathbf{0}$, otherwise the range of $f(\mathbf{x})$ is $\mathbb{R}$ because we assumed in the definition of the environments that $\Sigma_c^i \succ 0$. Furthermore, $\mathbf{w}_c = \mathbf{0}$ cannot be calibrated if $\mathbb{E}[Y \mid E = e_i]$ is not constant across environments; which is also part of the non-degeneracy constraints we required. Next we demand the equality of the intercepts across environments. Taking these equations and replacing Equation (12) into each of them, we get:

$$\mathbf{w}_c^{*\top}\mu_i^c - t\left(\mathbf{w}_c^\top\mu_i^c + (\mathbf{w}_{sp}^\top\mu_i)(\mathbf{w}_c^{*\top}\mu_i^c)\right) = \mathbf{w}_c^{*\top}\mu_j^c - t\left(\mathbf{w}_c^\top\mu_j^c + (\mathbf{w}_{sp}^\top\mu_j)(\mathbf{w}_c^{*\top}\mu_j^c)\right) \; \forall i,j \in [k].$$

Dividing both sides by $t$ and defining $\bar{\mathbf{w}}_c = \frac{\mathbf{w}_c^*}{t} - \mathbf{w}_c$, we can introduce another variable $t_2 \in \mathbb{R}$ and write this as a linear system of equations in variables $\mathbf{w}_{sp}, \bar{\mathbf{w}}_c, t_2$:

$$\bar{\mathbf{w}}_c^\top\mu_i^c - \mathbf{w}_{sp}^\top\mu_i(\mathbf{w}_c^{*\top}\mu_i^c) + t_2 = 0 \quad \forall i \in [k]. \tag{13}$$

We see that given $d_c + d_{sp} + 1$ environments, then with mild conditions on their non-degeneracy (i.e. the vectors containing the environment means and an extra entry of 1 span $\mathbb{R}^{d_c+d_{sp}+1}$), the only solution to the system is $\bar{\mathbf{w}}_c = 0, \mathbf{w}_{sp} = 0$, proving the last part of our statement.

Moving forward to demand multiple calibration on second moments $\mathbb{E}[Y^2 \mid f(X)_1 = \alpha, E = e_i] = \mathbb{E}[Y^2 \mid f(X)_1 = \alpha, E = e_j]$ for all $i, j \in [k]$, we may write this as:

$$\sigma_{y,i}^2 - \frac{\sigma_{f,y,i}^2}{\sigma_{f,i}^2} = \sigma_{y,j}^2 - \frac{\sigma_{f,y,j}^2}{\sigma_{f,j}^2} \quad \forall i,j \in [k].$$

Plugging Equation (12) into the above, a simplified expression is obtained:

$$\sigma_{y,i}^2 - t\sigma_{f,y,i} = \sigma_{y,j}^2 - t\sigma_{f,y,j} \quad \forall i,j \in [k].$$

Again we can divide by $t$ and obtain an explicit expression using $\bar{\mathbf{w}}_c, \mathbf{w}_{sp}$:

$$\bar{\mathbf{w}}_c^\top\Sigma_i^c\mathbf{w}_c^* - (\mathbf{w}_c^{*\top}\Sigma_i^c\mathbf{w}_c^* + \sigma_y)\mathbf{w}_{sp}^\top\mu_i = \bar{\mathbf{w}}_c^\top\Sigma_j^c\mathbf{w}_c^* - (\mathbf{w}_c^{*\top}\Sigma_j^c\mathbf{w}_c^* + \sigma_y)\mathbf{w}_{sp}^\top\mu_j \quad \forall i,j \in [k].$$

Finally, we can plug in Equation (13) and introduce another variable $t_3 \in \mathbb{R}$ to turn the above equations into:

$$\bar{\mathbf{w}}_c^\top\left(\Sigma_i^c\mathbf{w}_c^* + \left(\frac{\mathbf{w}_c^{*\top}\Sigma_i^c\mathbf{w}_c^* + \sigma_y^2}{\mathbf{w}_c^{*\top}\mu_i^c}\right)\mu_i^c\right) + t_2\left(\frac{\mathbf{w}_c^{*\top}\Sigma_i\mathbf{w}_c^*}{\mathbf{w}_c^{*\top}\mu_i^c}\right) + t_3 = 0.$$

It is now easy to see that if $k > d_c + 2$ and $\mathbf{M}_2(\{\mu_i, \Sigma_i\}_{i=1}^k, \sigma_y^2, \mathbf{w}_c^*)$ has full rank, the only solution to these equations satisfies $\bar{\mathbf{w}}_c = \mathbf{0}, t_2 = t_3 = 0$. When this is plugged into Equation (13), we find that if $k > d_{sp}$ and the spurious means span $\mathbb{R}^{d_{sp}}$ then the only possible solution is $\mathbf{w}_{sp} = \mathbf{0}$. Finally, $\bar{\mathbf{w}}_c = \mathbf{0}$ means $\mathbf{w}_c^* = t\mathbf{w}_c$, and if $f(\mathbf{x})$ is calibrated then we must have $t = 1$ since otherwise its estimate of the conditional mean is incorrect. Hence our proof is concluded. $\qquad\square$

We note that even though the setting we considered is restricted to causal features, anti-causal non-spurious features as those in Figure S1 can also be treated (resulting in the graph given in Figure 1). This is since for a single environment, the distribution $P[X_c, X_{\text{ac-ns}}, X_{\text{ac-spurious}}, Y \mid E = e]$ (we shorten here to $P^e$ for convenience) can always be written as follows, treating $X_{\text{ac-ns}}$ as causal features:

$$P^e[X_c, X_{\text{ac-ns}}, X_{\text{ac-sp}}, Y] = P^e(X_c, X_{\text{ac-ns}})P^e(Y \mid X_c, X_{\text{ac-ns}})P^e(X_{\text{ac-sp}} \mid Y, X_{\text{ac-ns}}, X_c)$$
$$= P^e(X_c, X_{\text{ac-ns}})P^e(Y \mid X_c, X_{\text{ac-ns}})P^e(X_{\text{ac-sp}} \mid X_{\text{ac-ns}}).$$

The last equality is due to the separation properties of the graph, and since the joint distribution is a multivariate Gaussian, so are all the factors in the above product. Hence each environment can be described using a structural equation model of the same type as Equation (11) and Theorem 2 applies.

# B  Dataset Statistics and Models

For each of the four WILDS experiments presented in Section 6, we briefly describe the data and report the splits we use for training, validation and test. In each experiment we train a model on the training set, and the calibrators on the validation set. The post-processing calibrators receive tuples of model predictions and labels as input, whereas fine tuning with CLOvE receives a latent representation (values of the last hidden layer for *Camelyon17* and *FMoW*, and average of the representation of the cls token over the last $4$ hidden layers in *CivilComments*). CLOvE is trained over a Multilayer Perceptron with 3 hidden layers, with batch size of $64$ and the Adam optimizer. We then compare all alternatives (Original, Naive Calibration, Robust Calibration and CLOvE) on the held-out test set (OOD). Whenever an In-Domain (ID) test set is available (*PovertyMap* and *Camelyon17*), we evaluate the model on it as well. Throughout our experiments, we measure and report the Expected Calibration Error (ECE) using 10 bins, dividing the $[0, 1]$ interval into sub-intervals of equal length. The licenses to the datasets are CC0 for *Camelyon17* and *CivilComments*, *FMoW* is distributed under the FMoW Challenge Public License and *PovertyMap* is public domain. All model training is done on an infrastructure with 4 RTX 2080 Ti GPUs.

## B.1  *PovertyMap*

**Problem Setting** *PovertyMap* is a regression task of poverty mapping across countries. Input $\mathbf{x}$ is a multispectral satellite image, output $y$ is a real-valued asset wealth index and domain $d$ is a country and whether the satellite image is of an urban or a rural area. The goal is to generalize across countries and demonstrate subpopulation performance across urban and rural areas.

**Data** *PovertyMap* is based on a dataset collected by [49], which organized satellite images and survey data from 23 African countries between 2009 and 2016. There are 23 countries, and every location is classified as either urban or rural. Each example includes the survey year, and its urban/rural classification.

1. Training: 10000 images from 13 countries.
2. Validation (OOD): 4000 images from 5 different countries (distinct from training and test (OOD) countries).
3. Test (OOD): 4000 images from 5 different countries (distinct from training and validation (OOD) countries).
4. Validation (ID): 1000 images from the same 13 countries in the training set.
5. Test (ID): 1000 images from the same 13 countries in the training set.

## B.2  *Camelyon17*

**Problem Setting** *Camelyon17* is a tumor identification task across different hospitals. Input $\mathbf{x}$ is an histopathological image, label $y$ is a binary indicator of whether the central region contains any tumor tissue and domain $d$ is an integer identifying the hospital. The training and validation sets include the same four hospitals, and the goal is to generalize to an unseen fifth hospital. We note that in [23] they include data from three hospitals in the training set and validate on data from a fourth hospital. Our setting includes a validation set from multiple hospitals since our fine tuning methods requires multiple domains.

**Data** The dataset comprises 450000 patches extracted from 50 whole-slide images (WSIs) of breast cancer metastases in lymph node sections, with 10 WSIs from each of five hospitals in the Netherlands [2]. Each WSI was manually annotated with tumor regions by pathologists, and the resulting segmentation masks were used to determine the labels for each patch. Data is split according to the hospital from which patches were taken.

1. Training: 335996 patches taken from each of the 4 hospitals in the training set.
2. Validation: 60000 patches taken from each of the 4 hospitals in the training set (15000 patches from each hospital).
3. Test (OOD): 85054 patches taken from the 5th hospital, which was chosen because its patches were the most visually distinctive.

### B.3 *CivilComments*

**Problem Setting** *CivilComments* is a toxicity classification task across different demographic identities. Input $x$ is a comment on an online article, label $y$ indicates if it is toxic, and domain $d$ is a one-hot vector with 8 dimensions corresponding to whether the comment mentions either of the 8 demographic identities *male*, *female*, *LGBTQ*, *Christian*, *Muslim*, *other religions*, *Black*, and *White*. The goal is to do well across all subpopulations, as computed through the average and worst case model performance.

**Data** *CivilComments* comprises 450000 comments, annotated for toxicity and demographic mentions by multiple crowdworkers, where toxicity classification is modeled as a binary task [4]. Each comment was originally made on an online article. Articles are randomly partitioned into disjoint training, validation, and test splits, and then formed the corresponding datasets by taking all comments on the articles in those splits.

1. Training: 269038 comments.
2. Validation: 45180 comments.
3. Test: 133782 comments.

### B.4 *FMoW*

**Problem Setting** *FMoW* is a building and land multi-class classification task across regions and years. Input $x$ is an RGB satellite image, label $y$ is one of 62 building or land use categories, and domain $d$ is the time the image was taken and the geographical region it captures. The goal is to generalize across time, and improve subpopulation performance across all regions.

**Data** *FMoW* is based on the Functional Map of the World dataset [6], which includes over 1 million high-resolution satellite images from over 200 countries, based on the functional purpose of the buildings or land in the image, over the years 2002–2018. We use a subset of this data introduced in [23], which is split into three time range domains, 2002–2013, 2013–2016, and 2016–2018, as well as five geographical regions as subpopulations: *Africa*, *Americas*, *Oceania*, *Asia* and *Europe*.

1. Training: 76863 images from the years 2002–2013.
2. Validation (OOD): 19915 images from the years from 2013–2016.
3. Test (OOD): 22108 images from the years from 2016–2018.
4. Validation (ID): 11483 images from the years from 2002–2013.
5. Test (ID): 11327 images from the years from 2002–2013.

**Models** In the following we briefly describe each of the models used in the experiments reported in Section 6.

- **BERT** - BERT is a 12-layer Transformer model [46] that represents textual inputs contextually and sequentially [9]. It is widely used in NLP, and is considered the standard benchmark for any state-of-the-art system. It was previously shown to be miscalibrated across its training and test environments [8]. In our *CivilComments* experiments, we use BERT-base-uncased, a smaller variant of BERT which has a layer size of 768
- **DenseNet** - Dense Convolutional Network (DenseNet), is a feed-forward neural network where for each layer, the feature-maps of all preceding layers are used as inputs, and its own feature-maps are used as inputs into all subsequent layers [19]. DenseNets are widely used in computer vision, especially for image classification tasks . We use a DenseNet-121 model, a DenseNet variant with 121 layers, in the *Camelyon17* and *FMoW* experiments.
- **ResNet** - Residual Network (ResNet) is a feed-forward neural network where layers are reformulated to learning residual functions with reference to the layer inputs [15]. DenseNets where shown to be successful in multiple image recognition tasks. We use the 18-layer variant, ResNet-18, in the *PovertyMap* experiment.

We run our models using the default setting used in [23]. Each model is trained four times, using a different random seed at each run. We report performance averages and their standard deviation in Section 6.

**Robustness to Model Architecture Choice**  For each of the five WILDS datasets we report results on (*PovertyMap*, *Camelyon17*, *CivilComments* and *FMoW*) also tested the robustness of our results to different model architectures. In the following we describe the architecture we tested for each dataset, and the relative results achieved.

- **BERT** - We used a pre-trained *BERT* in the *Civilcomments* experiments. On the *Civilcomments* dataset, we compared results on the BERT-base-uncased model with the cased and large versions. While we did find the performance increases with model size, perfromance drops on OOD examples remained consistent across models, with CLOvE outperforming *Robust Calibration* and *Naive Calibration* by an average of $1.4\%$ and $3.1\%$ (absolute), respectively.

- **DenseNet** - In the *FMoW* experiments, we tested the relative performance of the $121$ layer version to the $169$ and $201$ layer alternatives available via `https://pytorch.org/hub/pytorch_vision_densenet/`. Differences between the three models were not statistically significant.

- **ResNet** - In the *PovertyMap* experiments, we compare *ResNet-18* to the $34$ and $50$ layers alternatives available via `https://pytorch.org/hub/pytorch_vision_resnet/`. We found that *ResNet-18* performs slightly on the OOD test set, with average gain of $0.01$ in pearson correlation compared with *ResNet-34*. *Robust Calibration* remained better than *Naive Calibration* and the original model across runs.

**Training Algorithms**  In the WILDS experiments, for each dataset we train our models using three out of these four alternatives:

- **ERM** - Empirical risk minimization (ERM) is a training algorithms the looks for models that minimize the average training loss, regardless of the training environment.
- **IRM** Invariant risk minimization (IRM) [1] is a training algorithm that penalizes feature distributions that have different optimal linear classifiers for each environment.
- **DeepCORAL** DeepCORAL [44] is an algorithm that penalizes differences in the means and covariances of the feature distributions for each training environment. It was originally proposed in the context of domain adaptation, and has been subsequently adapted for domain generalization [11].
- **GroupDRO** - Group DRO [18] uses distributionally robust optimization (DRO) to explicitly minimize the loss on the worst-case environment.

We do not perform any hyperparameter search, and use the default version available in [23].

## C   Experiments on Colored MNIST

For the colored MNIST[8] dataset we trained Multi-Layer Perceptrons (MLPs) with ERM, IRMv1 and CLOvE, based on the code provided in [20] with the following adjustments: we add CLOvE and optimize it using SGD with batches of size $512$ from each training environment, for $5001$ steps at each run ( $50$ epochs). We used either the Adagrad optimizer [10] or Adam [22] (Adam was replaced with Adagrad in one environment where it produced highly unstable training metrics). All models were trained on a single NVidia Tesla P100 GPU virtual machine, on the Google Cloud Platform. Other algorithms were trained with Gradient Descent (i.e. without batching the dataset, which is infeasible for CLOvE since it is based on kernels) and Adam for $500$ steps/epochs, exactly as done in the code provided by [1, 20]. For CLOvE, hyperparamters are drawn similarly to the rest of the algorithms, except when using Adagrad where we multiply the originally drawn learning rate by $5$.

### C.1   Performance of CLOvE

We will refer to environments with tuples $(\alpha, \beta)$ that denote correlation with digit and color respectively, as done in Section 6.1. For each setting of training and test environments we experiment with, $100$ models are trained using each algorithm: ERM, IRM and CLOvE. To illustrate the failure

---

[8]The MNIST dataset is available under the terms of the Creative Commons Attribution-Share Alike 3.0 license

case pointed out in [20] and Section 6 of the paper, we train the algorithms with training environments corresponding to $e_1 = (0.1, 0.05), e_2 = (0.2, 0.05)$ and use data from test environment $e_3 = (0.9, 0.05)$. Figure S3 which we produce using code provided in [20] shows the results, where each point corresponds to a model trained with some set of drawn hyperparameters. Most models trained by CLOvE achieve log-loss that is close to that of the optimal invariant classifier (marked by dashed black line), while the models trained with IRMv1 are more scattered and specifically those that achieve lower log-loss are the ones that also obtain lower training objective. The bold colored lines mark the points that minimize $\sum_{e \in E_{train}} l^e(f_\theta) + \lambda \cdot r^e(f_\theta)$ with $\lambda = 10^6$ (expect for ERM where it's the point which minimizes the empirical loss), showing that out of the models trained with IRMv1, the one which minimizes the objective has loss close to that of the solution $OPT_{IRMv1}$ from Figure 3(a) in the paper (marked by dashed red line). That is while the CLOvE model with the lowest training objective is very close to the optimal invariant classifier in its test loss (marked by black dashed line). Note that in this case color is the invariant feature while the digit is spurious.

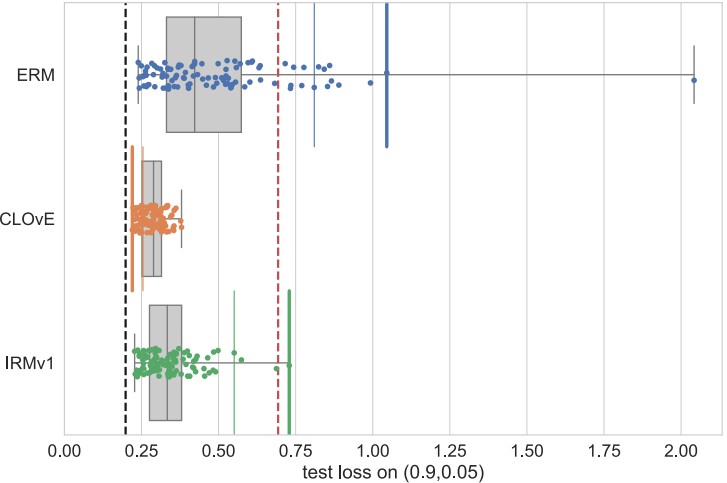

Figure S3: Log-loss on test environment $(0.9, 0.05)$ of classifiers trained with ERM, CLOvE and IRMv1 on training environments $(0.1, 0.05), (0.2, 0.05)$. Black dashed line marks the log-loss achieved by the optimal invariant classifier, while the red dashed line shows the loss achieved by $OPT_{IRMv1}$ from Figure 3(a). Bold colored lines mark the test loss achieved by the model which minimizes $\sum_{e \in E_{train}} l^e(f_\theta) + \lambda \cdot r^e(f_\theta)$ with $\lambda = 10^6$ out of all trained models.

For the opposite case, where the digit is invariant, the error incurred by MLPs in digit recognition makes it difficult to find the exact invariant classifier by optimizing CLOvE (since this error is close to the magnitude of the 0.05 correlation). Yet in Section C.2 the failure case of IRMv1 in these environments will be illustrated by average ECE (which CLOvE is a surrogate for) being a better measure of invariance than the IRMv1 objective.

The experiment presented in [1] used the training environments $e_1 = (0.25, 0.1), e_2 = (0.25, 0.2)$ with test environments $e_3 = (0.25, 0.9)$, where IRMv1 can in principle learn the optimal invariant classifier. We give the results on learning with these environments for completion. As can be observed in Figure S4, both CLOvE and IRMv1 learn models that are close to the optimal invariant one. While IRMv1 learned more of those models during the hyperparameter sweep[9], CLOvE still obtains some close-to-invariant models during the sweep. The rest of this section will be dedicated to studying model selection with the proposed average ECE criterion and the correlation between ID average ECE and OOD performance.

## C.2 Model Selection Experiments

Let us recall and elaborate the selection procedure proposed in Section 5:

---

[9]This can be attributed to the choice of ranges for drawing hyperparameters which we did not carefully tune to accommodate CLOvE.

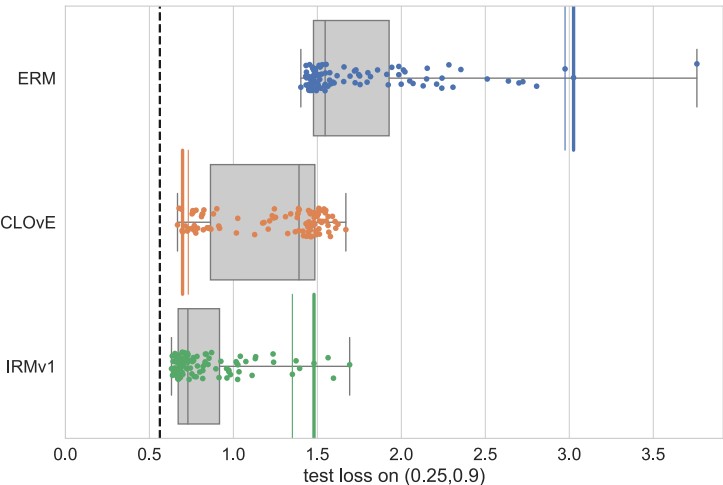

Figure S4: Log-loss on test environment $(0.25, 0.9)$ of classifiers trained with ERM, CLOvE and IRMv1 on training environments $(0.25, 0.1), (0.25, 0.2)$. Lines denote the same corresponding quantities in Figure S3, except we omit the red dashed line from that figure.

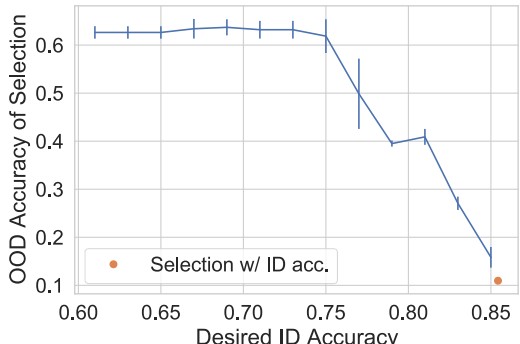

Figure S5: Model selection under a constraint on the ID accuracy. The OOD accuracy obtained by the proposed model selection method is plotted against the desired In-Domain accuracy, $\text{acc}_{\text{ID}}$, which is the minimal validation accuracy that we constrain the selected model to achieve. Red marker denotes the performance of the model achieved by selection based on ID validation accuracy alone.

- Given a desired threshold for In-Domain accuracy $\text{Thr}_{\text{ID}}$ and a set of models $f_1(\mathbf{x}), \ldots, f_n(\mathbf{x})$ from which we would like to select a candidate, perform the following.

- For each candidate model $\hat{f}$, recalibrate it with Isotonic Regression or some other preferred post-processing technique [10]. Calculate its ID validation error $\text{val}_{\text{ID}}(\hat{f})$ over a held-out dataset. For the held-out dataset from each environment $e \in E_{\text{train}}$ also calculate $ECE^e(\hat{f})$: the $ECE$ of $\hat{f}$ over this dataset. Then take $ECE(\hat{f}) = \sum_{e \in E_{\text{train}}} ECE^e(\hat{f})$.

- Choose $\arg\min_{\hat{f}: \text{val}_{\text{ID}}(\hat{f}) \geq \text{Thr}_{\text{ID}}} ECE(\hat{f})$.

**Selection with minimal ECE facilitates a tradeoff between ID accuracy and stability.** We use the trained models from the last section (all models trained with either ERM, IRMv1 or CLOvE are pooled into a set of candidates), over environments $e_1 = (0.25, 0.1), e_2 = (0.25, 0.2)$. Selecting the model with minimal $\text{val}_{ID}(\hat{f})$ delivers a classifier with $10.96\%(\pm0.81)$ accuracy on $e_{\text{test}} = (0.25, 0.9)$ and $85.43\%(\pm0.13)$ accuracy on the training environments. The trade-off achieved by selection with

---

[10]This is a crucial step, since models that are highly miscalibrated can become well-calibrated upon post-processing

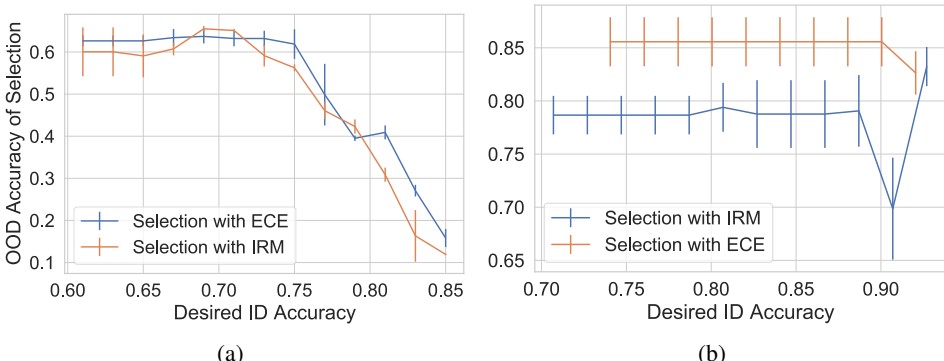

(a)                                    (b)

Figure S6: Comparison of proposed model selection procedure applied with IRMv1 objective and the average ECE over training environments in two settings. (a) $e_1 = (0.05, 0.1), e_2 = (0.05, 0.2), e_{\text{test}} = (0.05, 0.9)$ and (b) $e_1 = (0.25, 0.1), e_2 = (0.25, 0.2), e_{\text{test}} = (0.25, 0.9)$.

the proposed criterion is shown in Figure S5. Demanding ID accuracy that is higher than $75\%$ (the ID error obtained by an optimal invariant classifier) yields a relatively sharp drop towards the OOD accuracy obtained by a classifier that purely minimizes empirical error. Going below $75\%$ retrieves a classifier that achieves $64.98\%(\pm2.67)\%$ OOD accuracy.

**Comparison with IRMv1 Penalty as Selection Criterion.** As a baseline to the average ECE over training environments we compare it with using the value of the IRMv1 regularizer, also calculated with a validation set from each training environment. In Figure S6 we compare the curves obtained by the proposed model selection procedure, and that same procedure when replacing the ECE with the value of IRMv1. Figure S6(a) shows the result on the scenario where $e_1 = (0.25, 0.1), e_2 = (0.25, 0.2)$ and $e_{\text{test}} = (0.25, 0.9)$. In this case the two methods are quite comparable, expect for the tail of high desired ID accuracies, where the chosen models are trained with ERM and the IRMv1 criterion fails to rank them by their OOD accuracy. Figure S6(b) shows the same plot on the scenario where $e_1 = (0.05, 0.1), e_2 = (0.05, 0.2)$ and $e_{\text{test}} = (0.05, 0.9)$, which corresponds to the failure case of IRM in Figure 3(a). Due the observation of [20], we may expect the IRMv1 objective to fail at capturing invariance in this setting. Indeed, the model selection done using the IRMv1 penalty gives a worst model than the one selected by ECE in this case. In Figure S7 we also plot the correspondence between OOD accuracy and these quantities (namely ID average ECE, and IRMv1 penalty) as in Figure 3(b) for both settings depicted in Figure S6 showing the erratic behavior of the IRM penalty when considered on different training regimes.

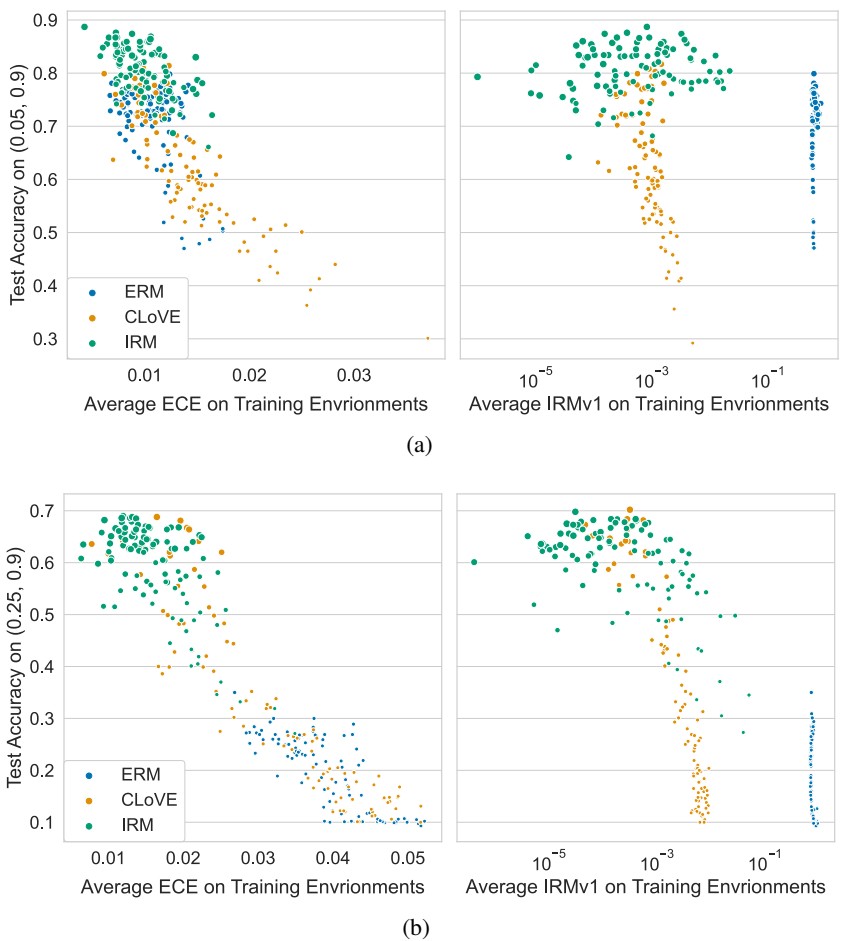

Figure S7: Scatter plots of average ECE, and average IRMv1 penalty over training environments against the accuracy on test environments in settings (a) $e_1 = (0.05, 0.1), e_2 = (0.05, 0.2), e_{\text{test}} = (0.05, 0.9)$ and (b) $e_1 = (0.25, 0.1), e_2 = (0.25, 0.2), e_{\text{test}} = (0.25, 0.9)$. Size of marker is proportional to the ratio between OOD and ID accuracies.