# OpenReview forum: "On Calibration and Out-of-Domain Generalization"
_NeurIPS.cc/2021/Conference — NeurIPS 2021 Poster_

### Official Review · Reviewer_jGwe · 2021-07-05

**Rating:** 7
**Confidence:** 3

**Summary:**

This work suggests that for well calibrated models will achieve better performance on OOD data. It is shown empirically that expected calibration error (ECE) is negatively correlated with OOD performance, and various methods of selecting calibrated models are suggested. The best performing method involves regulation to encourage calibration during training.

**Limitations And Societal Impact:**

Adequately addressed.

**Main Review:**

This paper is clearly written and explores interesting ideas. The naive and rob. calibration methods make excellent baselines, and the empirical performance gain, in particular for FMoW/Camelyon17, is impressive. Limitations are addressed, in particular that only fine-tuning is explored and that theoretical findings are limited to the linear setting. A few questions:

- Are these models still calibrated out of distribution? This would strengthen the argument.
- Why does this work only consider OOD performance. Could this be a method to improve the calibration of models during training instead of needing to apply post-processing? Does it result in lower ECE than standard post processing, or could this method negatively impact accuracy?
- As I am less familiar with these ideas, I would have appreciated additional intuition. I did not find the motivating hospital example particularly obvious/insightful. The following motivation makes more sense to me: If I have two classifiers which have similar accuracy but one is more calibrated, I would rather choose the calibrated classifier as I would assume it is using the right information. However, I am not sure why multi-domain is important for this motivation or method?

-- Added --

I acknowledge the response and will keep my score as is

**Time Spent Reviewing:**

2.5h

---

> ### Author Response · Authors · 2021-08-10
> **Response to Reviewer 4**
>
> We thank the reviewer for their comments and support of the paper.
>
> > Are these models still calibrated out of distribution? This would strengthen the argument.
>
> Yes, when model performance increases on OOD examples so does the model’s calibration on OOD examples improve. In fact, the relationship between OOD calibration and performance is quite strong in our experiments. In the revised version of the paper we will add the OOD calibration results.
>
> > Why does this work only consider OOD performance. Could this be a method to improve the calibration of models during training instead of needing to apply post-processing? Does it result in lower ECE than standard post processing, or could this method negatively impact accuracy?
>
> The work we build upon, i.e. [1], is aimed at improving calibration and is already effective in that sense. Therefore we focus on the novel aspect of our work, which is the connection between calibration on multiple training domains and OOD performance.
>
> In our setting, fine-tuning with the proposed loss function often indeed gave better ECE than standard post-processing (e.g. Isotonic Regression in the paper). The values on the x-axis of Figure 4 in the paper show this quantitatively.
>
> > I would have appreciated additional intuition
>
> The reviewer is indeed correct when saying that, everything else being equal, a calibrated classifier is better than an uncalibrated one. In our work we show that furthermore, if a classifier is well-calibrated across a diverse set of domains, it automatically learns to ignore any spurious features that might exist. This is different from normal accuracy, as it guarantees that even on new, unseen domains, these spurious correlations will not be used.
>
> How does this come about? Intuitively, relying on a spurious feature during training might increase accuracy in certain domains, but will destroy the simultaneous calibration because by definition a spurious correlation is not constant across domains.
>
> [1] Kumar, A., Sarawagi, S., & Jain, U. (2018, July). Trainable calibration measures for neural networks from kernel mean embeddings. In International Conference on Machine Learning(pp. 2805-2814). PMLR.

---

### Official Review · Reviewer_QH4q · 2021-07-15

**Rating:** 7
**Confidence:** 3

**Summary:**

This paper draws an important connection between out-of-domain generalization and calibrated models. Specifically, it is argued that calibrated models are related to invariant representations w.r.t spuriously correlated features, which is necessary for generalization across multiple domains. The paper proves this statement in the case of a multivariate linear-Gaussian model. The authors empirically find a correlation between the Expected Calibration Error (ECE) and OOD performance on Colored MNNIST. Based on these insights, the authors propose three methods for achieving multi-domain calibration of trained models in practice: Approach #1 suggests doing model selection based on the Expected Calibration Error; Approach #2 is intended to augment the trained model by post-processing calibration methods and Approach #3 introduces a calibration objective at train time. In multiple experiments across 5 datasets, reported results suggest that their method outperforms the baselines algorithms without calibration on OOD test environments.


**Limitations And Societal Impact:**

There is a very brief discussion on limitations in section 7, which could be discussed more prominently in the respective sections. Societal Impact is addressed properly.

**Main Review:**

**Originality:** To the best of my knowledge this work is an original and novel contribution in relating the problem of finding invariant representations for out-of-domain generalization with the field of calibrated models. It offers important insights, both theoretically and practically and proposes approaches that are agnostic to well-known baseline algorithms in OOD generalization. The related work seems to be adequately covered but I am not familiar enough with the entire literature.

**Quality:** This paper seems technically correct and the claims are well supported and motivated by theorems in the case of a multivariate linear gaussian model. The proposed methods appear to be the effective approaches once the link between calibration and OOD generalization has been established. I did not find any technical flaws in the experiments both on Coloured MNIST and the four chosen datasets from the WILDS Benchmark. Analysis and discussion of the experimental results are adequate. The theoretical claims are somewhat limited in that they are constrained to linear models but the authors are upfront about this and the experiments support that calibration still offers benefits beyond linear settings.

**Clarity:** I found most parts of this work clearly written and well organized.

I had some problems parsing section 5.3 and how the CLOvE method works in detail. From reading this part, I assumed this method trains the models from scratch under the proposed CLOvE regularizer but in the experiments you only applied it as a fine-tuning method, which confused me because training from scratch seems to be the natural first choice to test this method. What was the reason for this particular choice of only fine tuning the top layers?

Additionally, I think the paper could benefit if the assumed graph and problem setting in 2.1 is supported or accompanied with a concrete simple example, as with the hospital example in the introduction. This could help to get some intuition about what could be a spurious or non-spurious feature and why we would like to avoid them.

**Significance:** I can imagine results in this paper are clearly important from a theoretical and practical point of view as this work establishes an interesting link with a seemingly unrelated field. While the proposed method already shows improvements upon existing algorithms this opens up a path for further research w.r.t to theory and improved methods.

----
Update: I want to thank the authors for their response and running the additional experiments. I also read the other reviews and remain very positive about this work and will therefore maintain my score.

**Time Spent Reviewing:**

10

---

> ### Author Response · Authors · 2021-08-10
> **Response to Reviewer 3**
>
> We thank the reviewer for their careful review and important comments.
>
> > ... What was the reason for this particular choice of only fine tuning the top layers?
>
> We thank the reviewer for raising this important issue. Following this comment we trained the models from scratch on Camelyon17, FMoW and CivilComments using CLOvE. From our preliminary results, we observe that the accuracies we achieve with the end-to-end models are comparable to those we get with fine-tuning, with an average performance of 0.75 (absolute) lower than the fine-tuned models (for example, end-to-end CLOvE achieves an accuracy of 75% on Camelyon17, compared with 75.75% it achieves when fine-tuning ERM with CLOvE); we will include the final results in the supplementary material.
>
> To clarify why we opted for fine-tuning in our experiments in the first place, we note that our main goal is to examine whether improving multi-domain calibration improves OOD generalization and to isolate its effect. We believe fine-tuning displays a good trade-off between the complexities introduced by training large models and the ability to achieve significant accuracy gains. Since fine-tuning and post-processing are also very common approaches for improving model calibration, as well as in general in NLP and computer vision, we found it suitable to base our experiments on them. We note that we did however train models from scratch in smaller datasets like Colored MNIST.
>
> > the paper could benefit if the assumed graph and problem setting in 2.1 is supported or accompanied with a concrete simple example, as with the hospital example in the introduction.
>
> We will add examples of such features to the illustrative case given in the third paragraph of Section 1. For example, when learning from health records we may consider E as different hospitals, $X_{\text{causal}}$ is smoking, which is a cause of $Y$ that indicates whether a patient has lung cancer, and then $X_{\text{ac-non-spurious}}$ may correspond to known symptoms of lung cancer such as chest infections that are manifested in a chest x-ray included in the health record, while $X_{\text{ac-spurious}}$ can be e.g. notes or marks that are written over these x-ray images as in [1]. Smoking habits may vary across populations, and the notes written over slides may differ according to common practice in different hospitals, but the influence of smoking on cancer and the manifestation of cancer in an x-ray might not vary by hospital.
>
> [1] Zech, J. R., Badgeley, M. A., Liu, M., Costa, A. B., Titano, J. J., & Oermann, E. K. (2018). Variable generalization performance of a deep learning model to detect pneumonia in chest radiographs: a cross-sectional study. PLoS medicine, 15(11), e1002683.

---

### Official Review · Reviewer_5YBQ · 2021-07-15

**Rating:** 7
**Confidence:** 3

**Summary:**

The paper proposes out-of-domain (OOD) generalization of the machine learning models by calibrating them across multiple domains. The paper argues that multi-domain calibration makes the model learn invariant features and therefore free of spurious correlations. The relationship between calibration and invariance has been clearly stated via lemma. The authors prove how multi-domain calibration makes the model free of spurious correlations in the case of a linear Gaussian model and in two different scenarios. Different ways of practically achieving multi-domain calibration, i.e., during model selection, as post-processing techniques on the trained model, and during training have been proposed. Experimental results on the WILDS OOD benchmark and Colored MNIST dataset illustrate the efficacy of the proposed ways for calibrating a model on multi-domains for OOD generalization.


**Ethical Concerns:**

The authors adhere to the ethical standards mentioned in the NeurIPS Ethics Guidelines.


**Limitations And Societal Impact:**

The motivation for OOD generalization in Line 23 of the introduction states that "a medical diagnosis system trained on patient data from a few hospitals could fail when deployed in a new hospital.".  OOD generalization might increase trust in the use of these models in the medical domain. The medical domain is one of the most safety-critical domains. The question to be asked here by these multi-domain calibrated models is: Are they good enough to be trusted in safety-critical domains?


**Main Review:**

Strengths:
1) All the definitions and lemmas are very well written.
2) The novel theoretical results on the relation between multi-domain calibration and spurious correlation are a promising step towards OOD generalization.
3) The paper is not restricted to the theory of why multi-domain calibration works for OOD generalization but also proposes practical ways of achieving it during model selection, as post-processing techniques on the trained model, or during training.
4) The experimental results are quite rich and good.

Weakness:
1) The paper is not so easy to follow due to the lack of background or citation(s) for the definitions of causal, anti-causal, spurious features, and isotonic regression, etc. Also, a small paragraph on the mathematical notations used throughout the paper would be helpful. Adding an exact reference to the appendix sections will be helpful to link the paper with the appendix.
2) The motivation for using the causal graph in Figure 1 of Section 2.1 (problem definition) is missing. Is it the standard causal graph used to describe the relationship between the types of features, environment, and the label? Also, the causal graphs used in Section 3 for proving that multi-domain calibrated models are free of spurious features are different from the one in Figure 1. Why did the authors split the graph used in defining the problem statement into two and then proved the properties for these two sub-graphs? The justification for doing so is missing in the paper. Is it possible to prove the desired properties on the entire graph from Figure 1?
3) The authors use invariance and free of spurious correlations interchangeably in the paper. A lemma for invariance <=> free of spurious correlations would be good.
4) Theorem 1 and 2 are proved for binary classifier and regression, respectively. Do both of these theorems hold for multi-class classifier and regression?
5) The authors mention that CLOvE is a calibration technique applicable during training in Line 247. But in experiments, they use CLOvE as a fine-tuning approach on the models already trained with the existing OOD generalization techniques. This has been acknowledged in conclusion (Line369) but it would be good to evaluate the effectiveness of CLOvE by using it in training models from scratch (not based on other OOD generalization techniques).
6) There is a disconnect between the theorems (1 and 2) on calibrated models and the proposed ways of calibrating models on multi-domains. Do the proposed way of calibrating models satisfy the conditions required in the theorems?

Minor comments:
1) All the definitions given in the paper are for a binary classifier. Are they generalizable for a multi-class classifier?
2) What is w* in line 111?
3) There is a notation mismatch for R^d_{sp} in lines 152 and 164.
4) 6 - 9 in line 278 should be 5 - 9.
5) What is table S1 in the appendix? It is not referred to anywhere.
6) It should be "Section 6" instead of "Section 7" in line 847 of the appendix.


**Time Spent Reviewing:**

~11.5 hours

---

> ### Author Response · Authors · 2021-08-10
> **Response to Reviewer 2 - main comments**
>
> We thank the reviewer for recognizing the strengths and novelty of the paper, and for raising concerns which we address below.
>
> > The paper is not so easy to follow due to the lack of background or citation(s)...
>
> We thank the reviewer for the comment, and agree that more background, especially around our use of causality-related terms would improve the paper. Following the reviewer’s point we will introduce the following changes to the manuscript:
> * On top of the references we included for Isotonic Regression, e.g. [1], we will give a short review of the method and a derivation of the robust variant that we propose. We will add this along with a definition of the ECE and a summary of relevant results for the MMCE regularizer, as a section in the supplementary material. A reference to this section will be added to the main paper.
> * Generally, all references to the supplementary material will be changed to include the appropriate section number.
> * Mathematical notations will be clarified in the first paragraph of Section 2.1.
> * Please see response to the next point regarding definition of $X_{\text{causal}}$ etc.
>
> > The motivation for using the causal graph in Figure 1 of Section 2.1 (problem definition) is missing. Is it...
>
> We agree that the motivation for the causal graph should also be better explained. The graph in Figure 1 depicts the main assumption that our analysis relies on, where the environment does not affect $Y$ directly. A similar graph appears in Figure 3.1 of [2], where we rename the variables and add $X_{\text{causal}}$. $X_{\text{causal}}$ denotes features that might undergo covariate shift, but $P(Y \mid X_{\text{causal}})$ is fixed across environments. Hence, the setting we consider is more general than the one in [2].
>
> To better describe the intuition behind the names $X_{\text{causal}}, X_{\text{ac-non-spurious}}$ and $X_{\text{ac-spurious}}$, we will add examples of such features to the illustrative case given in the third paragraph of Section 1. For example, when learning from health records, we may consider E as different hospitals, $X_{\text{causal}}$ is smoking, which is a cause of $Y$ that indicates whether a patient has lung cancer, and then $X_{\text{ac-non-spurious}}$ may correspond to known symptoms of lung cancer such as chest infections that are manifested in a chest x-ray included in the health record, while $X_{\text{ac-spurious}}$ can be notes or marks that are written over these x-ray images as in [6]. Smoking habits may vary across populations, and the notes written over slides may differ according to common practice in different hospitals, but the influence of smoking on cancer and the manifestation of cancer in an x-ray might not vary by hospital.
>
> > Why did the authors split the graph used in defining the problem statement into two?
>
> This is a good point. We initially thought that it would be clearer to split the analysis into cases. Theorem 1 which extends the analysis of [2] is quite natural to consider as a motivating example for the classification problems of interest. The case of causal features (Fig. 2(a) in the paper) lends itself to simpler analysis in the regression case, since it maintains the joint Gaussian distribution under any covariate shift.
> Following the reviewer’s comment, we looked back at how the analysis of the full Fig. 1 graph could work out. We find that we can in fact extend the graph used in Theorem 2 to include anti-causal non-spurious features (and thus coincide with the Fig. 1 graph), and still use the same proof technique of Theorem 2 by a change of variables and a straightforward reduction. We will therefore also include the extended result in the supplement of the paper. Thank you for spurring us into obtaining this fuller result.
>
> > The authors use invariance and free of spurious correlations interchangeably in the paper.
>
> The reviewer is correct in pointing this out, and we regret any confusion this might have caused. Our interchangeable use of “free of spurious correlations” and “invariance” follows the similar use in IRM [7], where the authors distinguish between invariant correlations (i.e. those that are stable across all environments) and spurious ones (those that are not). Following their use, we indeed define the two notions as equivalent in line 75, where we define a representation $\Phi(X)$ to contain a spurious correlation if $Y \not\perp E \mid \Phi(X)$ (i.e. if it is not invariant).
> Following the reviewer’s comment we will clarify and emphasize this equivalence, and its relation to the nomenclature used in previous work.
>
> > Theorem 1 and 2 are proved for binary classifier and regression, respectively. Do both of these theorems hold for multi-class classifier and regression?
>
> For Theorem 1, it is possible to change the setting in a manner that suits multi-class prediction. One simple extension is where $P(X | Y=k)$ is Gaussian with a different mean for each class. Upon using a suitable definition of multi-class calibration (e.g. “distribution calibration” in [4], which relies on the definition of [5]), a similar derivation to the one provided in our proof will hold. That is, calibration will correspond to a set of quadratic constraints, which under a diverse set of environments can only be satisfied if $w_{\text{ac-spurious}}$ is 0.
>
> As for Theorem 2, we assume that by multi-class the reviewer refers to regression with multiple labels/targets. Then as often occurs in multi-target linear regression, we may apply the analysis to each target separately and arrive at the desired result.
>
> > The authors mention that CLOvE is a calibration technique applicable during training in Line 247. But in experiments, they use CLOvE as a fine-tuning approach...
>
> We thank the reviewer for raising this important issue. Following this comment we trained the models from scratch on Camelyon17, FMoW and CivilComments using CLOvE. From our preliminary results, we observe that the accuracies we achieve with the end-to-end models are comparable to those we get with fine-tuning, with an average performance of 0.75 (absolute) lower than the fine-tuned models (for example, end-to-end CLOvE achieves an accuracy of 75% on Camelyon17, compared with 75.75% it achieves when fine-tuning ERM with CLOvE); we will include the final results in the supplementary material.
>
> To clarify why we opted for fine-tuning in our experiments in the first place, we note that our main goal is to examine whether improving multi-domain calibration improves OOD generalization and to isolate its effect. We believe fine-tuning displays a good trade-off between the complexities introduced by training large models and the ability to achieve significant accuracy gains. Since fine-tuning and post-processing are also very common approaches for improving model calibration, as well as in general in NLP and computer vision, we found it suitable to base our experiments on them. We note that we did however train models from scratch in smaller datasets like Colored MNIST.
>
> > There is a disconnect between the theorems (1 and 2) on calibrated models and the proposed ways of calibrating models on multi-domains. Do the proposed way of calibrating models satisfy the conditions required in the theorems?
>
> The theorems portray the guarantees of perfect multi-domain calibration under an ideal case without finite sample limitations. Then to examine what happens in real-world datasets and practical calibration techniques, we used:
> * The Expected Calibration Error (ECE), which is standard for evaluating the calibration of models (e.g. [3]). The average ECE and the proposed regularizer (CLOvE) are consistent with Theorem 1 in the sense that they equal 0 if and only if the conditions of the theorem are met.
> * Isotonic Regression, a common post-processing method that is known to improve calibration in practice. The connection between vanilla isotonic regression, which we referred to as “naive calibration”, and the conditions in Theorem 1 is not straightforward, since it does not consider multiple environments. This is why as a next step we considered robust calibration, a simple adjustment that seeks calibration across environments.
>
> These methods display a varying level of connection to the theoretical results. We believe they allow us to better probe on the experimental front the practical implications of calibration, including methods that are very widely used by practitioners such as Isotonic Regression.
> Indeed, we can see in the experimental results that the calibration method that is most powerful and tightly connected to the goals set out in the theorems (i.e. CLOvE), gives the best OOD accuracy. Then robust calibration in most cases had the second best performance, followed by vanilla isotonic regression. We would be happy to take any suggestions on calibration methods that will better connect to the theoretical part. We will clarify the formal connections between the proposed methods and multi-domain calibration in the section that will be added to the appendix about isotonic regression and the ECE, along with appropriate references.

---

> ### Author Response · Authors · 2021-08-10
> **Response to Reviewer 2 - minor comments, limitations and references**
>
> **Minor Comments**
>
> > All the definitions given in the paper are for a binary classifier. Are they generalizable for a multi-class classifier?
>
> Generally yes, please see response regarding whether Theorem 1 can be extended to the multi-class case.
>
> > What is $w*$ in line 111?
>
> $w*$ is a function from $\mathcal{H}$ to $[0,1]$, a classifier on top of the representation. We indeed missed this definition, and thank the reviewer for pointing this out. We will better clarify the role of $w^*$, though a complete overview of IRM is out of scope for the main paper.
>
> > Typos and table S1
>
> We thank the reviewer for noting the mistakes in notations, they will be fixed. Regarding table S1, it presents results on Amazon Product Reviews, an additional dataset used in Wilds. As we already have results for a binary text classification task (CivilComments) in the main paper, and because domain adaptation results in NLP show that there is not much of an OOD problem on this dataset, we left it in the Appendix. We will add a description of the dataset and a discussion of the results in the revised version of the paper.
>
> **Comments regarding limitations and societal impact**
> > ...The question to be asked here by these multi-domain calibrated models is: Are they good enough to be trusted in safety-critical domains?
>
> We thank the reviewer for bringing up this crucial point. Proactive approaches to mitigate instabilities due to distribution shifts are a vital part of working towards safety in healthcare and other domains (e.g. [8], section 3 and references within). Calibration on multiple domains fits into these approaches as it attempts to mitigate such instabilities without observing data from the test domains. In terms of safety, we suggest that multi-domain calibration should be evaluated for safety-aware application, and if a model does not satisfy it then this is a cause for concern. Thus calibration should either be improved, or the source of miscalibration should be understood if the model is to be deployed in a critical setting.
>
> We do not claim that calibration across multiple domains is in any way sufficient, or should be the only means to ensure safety. For one, it does not guarantee good performance under all relevant dataset shifts (e.g. we still need diverse datasets to train on, and perhaps more advanced methods to enforce invariance). Multi-domain calibration is a useful tool and can open up paths for further improvements, but it is still not a full solution. Besides stability under dataset shifts, there are other facets of safety that multi-domain calibration does not address at all, e.g. recognizing failure identification and maintenance of machine learning systems [9]. We will extend and clarify the discussion of these points in our paper.
>
> **References**
>
> [1] Niculescu-Mizil, A., & Caruana, R. (2005, August). Predicting good probabilities with supervised learning. In Proceedings of the 22nd international conference on Machine learning (pp. 625-632).
>
> [2] Rosenfeld, E., Ravikumar, P., & Risteski, A. (2021). The Risks of Invariant Risk Minimization. In Proceedings of the 9th international Conference on Learning Representations.
>
> [3] Guo, C., Pleiss, G., Sun, Y., & Weinberger, K. Q. (2017, July). On calibration of modern neural networks. In International Conference on Machine Learning (pp. 1321-1330). PMLR.
>
> [4] Zhao, S., Kim, M. P., Sahoo, R., Ma, T., & Ermon, S. (2021). Calibrating Predictions to Decisions: A Novel Approach to Multi-Class Calibration. arXiv preprint arXiv:2107.05719.
>
> [5] Kull, M., & Flach, P. (2015, September). Novel decompositions of proper scoring rules for classification: Score adjustment as precursor to calibration. In Joint European Conference on Machine Learning and Knowledge Discovery in Databases(pp. 68-85). Springer, Cham.
>
> [6] Zech, J. R., Badgeley, M. A., Liu, M., Costa, A. B., Titano, J. J., & Oermann, E. K. (2018). Variable generalization performance of a deep learning model to detect pneumonia in chest radiographs: a cross-sectional study. PLoS medicine, 15(11), e1002683.
>
> [7] Arjovsky, M., Bottou, L., Gulrajani, I., & Lopez-Paz, D. (2019). Invariant risk minimization. arXiv preprint arXiv:1907.02893.
>
> [8] Saria, S., & Subbaswamy, A. (2019). Tutorial: Safe and Reliable Machine Learning. ArXiv, abs/1904.07204.
>
> [9] Subbaswamy, A., & Saria, S. (2020). From development to deployment: dataset shift, causality, and shift-stable models in health AI. Biostatistics, 21(2), 345-352.

---

> > ### Comment · Reviewer_5YBQ · 2021-08-21
> > **Feedback on the response**
> >
> > Thank you for all the clarifications and for performing additional experiments.
> >
> > The only concern that remains is about the explicit connection between the proposed "robust calibration" and the theoretical guarantees. Could you explain it more explicitly?

---

> > > ### Author Response · Authors · 2021-08-22
> > > **Connection between robust calibration and theoretical guarantees**
> > >
> > > Thank you for considering our response and raising the remaining important concern. Below, when referring to “robust calibration” we address both the proposed “Robust Isotonic Regression” method and CLOvE. We give a more detailed explanation about both below. The main connection between all the methods and the theoretical results is that whenever the proposed methods achieve 0 error then the classifier is simultaneously calibrated over all environments, thus satisfying the conditions of the theorems. We further elaborate on each method below, and would be happy to provide additional details if required.
> > >
> > > **Robust Isotonic Regression**: Before considering the robust variant, please note that one way we could interpret the benefit of Isotonic Regression towards better calibration is as follows: Isotonic Regression finds a monotonic function from $[0,1]$ to $[0,1]$ that minimizes the Brier score (i.e. the MSE applied to a binary classification problem), which in turn can be decomposed to the sum of a refinement and calibration scores; see e.g. [1, section 3.2] and references therein.
> > > Since the refinement score only depends on the order of the examples w.r.t the classifier outputs (see [2, section 3.1, fourth paragraph]), it is not affected by monotonic functions, up to subtleties regarding the binning of the $[0,1]$ interval. Isotonic Regression can then be seen as a method that minimizes a type of calibration error. Specifically, this calibration error equals 0 if and only if the model is calibrated. Yet Isotonic regression does not take the multiple environments into account, and could yield a classifier that is well calibrated on the entire dataset, pooled from all environments, but not on individual environments.
> > >
> > > For the robust version, we simply find a monotonic function which minimizes the worst-case Brier score over environments. Following the reasoning laid out above for the vanilla Isotonic Regression, it can be seen as minimizing the worst-case calibration error over the different environments. Thus it is better geared towards multi-environment calibration, e.g. the worst-case calibration error is 0 if and only if the classifier is calibrated over all environments, as required in the conditions of Theorem 1. However it has limited expressive power due to the constraint that the function must be monotonic, which is why we turned to consider CLOvE.
> > >
> > > **Calibration Loss Over Environments (CLOvE):** CLOvE’s key connection to the theoretical guarantees lies in Theorem 1 of [3], which states that the MMCE regularizer equals 0 if and only if the classifier is perfectly calibrated. Since our proposed regularizer sums MMCE across environments, we conclude that it equals 0 if and only if the classifier is calibrated on all environments, thus satisfying the requirements in our Theorem 1.
> > > Theorems 2 and 3 of [3] can also be adapted to the multi-environment setting; they provide guarantees on finite sample scenarios and on the quality of approximation to the ECE. Since our theory only concerns the case of infinite data, the connection of these theorems to our results is less explicit, and hence in our revised manuscript we will mention them in the appendix.
> > >
> > > [1] Kull, M., & Flach, P. (2015, September). Novel decompositions of proper scoring rules for classification: Score adjustment as precursor to calibration. In Joint European Conference on Machine Learning and Knowledge Discovery in Databases(pp. 68-85). Springer, Cham.
> > >
> > > [2] Bella, A., Ferri, C., Hernández-Orallo, J., & Ramírez-Quintana, M. J. (2013). On the effect of calibration in classifier combination. Applied intelligence, 38(4), 566-585.
> > >
> > > [3] Kumar, A., Sarawagi, S., & Jain, U. (2018, July). Trainable calibration measures for neural networks from kernel mean embeddings. In International Conference on Machine Learning (pp. 2805-2814). PMLR.

---

> > > > ### Comment · Reviewer_5YBQ · 2021-08-23
> > > > **Thanks**
> > > >
> > > > Thank you for clarifying the main concern. I increased my rating from 5 to 7.

---

### Official Review · Reviewer_C1X7 · 2021-07-16

**Rating:** 7
**Confidence:** 4

**Summary:**

The authors suggest a different regularizer than IRM based on the gradient  - which has been shown to have failure modes
They instead propose calibration and, focusing on classification (but also with some regression experiments),
(i) show for linear model that under similar conditions as the general position condition for IRM, for k > 2 d_sp a multi-domain calibrated w has zero coefficient on the spurious features
(ii) use MMCE for nonlinear setting on WILDS dataset that illustrate that their method works.

*Update:* I've carefully read the author's response that has addressed my comments and I still think it's a good paper and should be accepted.


**Limitations And Societal Impact:**

They could have a more thorough discussion about limitations (as proposed in detailed comments). Potential negative societal impact same as out-of-distribution detection in general.

**Main Review:**

Strengths
- Clearly and well written
- principled derivation for using multi-class calibration (in practice via the MMCE proxy), quick glance at the proof makes sense
- good experiments demonstrating impressive effectiveness of calibration on WILDs dataset, comparing also with some alternative calibration methods

Critical comments:

In terms of theory
- Theorem (that calibration works for sufficiently different training environments) is not surprising given IRM Theorem, uses similar technique -> don’t really see it as a really novel or groundbreaking contribution
- In contrast, more useful/convincing for their story would have been a theorem that shows where calibration works but IRM does not

In terms of methodological contribution
- the numbers in orig column for Camelyon 17 in Table 2 are not matching the WILDS - in the appendix you can find that the problem is in the choice of the training dataset, i.e. you are using 4 rather than 3 training environments. in the original WILDS paper with 3 training environments ERM had best OOD generalization. how does Clove+ERM compare with IRM for example, for different (i.e. smaller) number of training environments? how about other architectures?
- would be good to add a discussion about computational costs compared to other methods

Experimental details:
- need some more experimental details on how you use the clove regularizer in practice even though one can guess that you use the same settings as in Kumar et al, Laplace kernel …? it would greatly improve reproducibility (no code was given to us)
- how have the authors chosen the parameters for measuring ECE and for post-hoc calibration (e.g. number of bins)?
- what if you don’t use Clove as fine-tuning but e.g. the IRMv1 regularizer on the entire network when trained from scratch?
generally: are there scenarios in which post-hoc calibration fails to perform better than methods that exploit multi-domain information during training (e.g. GroupDRO, DeepCORAL, IRM)? what is characteristic for these failure cases? moreover, when does a decrease in ECE not imply an improvement in OOD accuracy?

Minor comments:
- important typo odd classifiers: f(1,-1) = - f(-1,1) - the minus was missing
- in Theorem 1, it would be much clearer to write x \in R^{d_{sp}} (apart from the fact that the subscript sp was a typo).
- in appendix d_e = d_sp
- the label set for binary classification is not consistent across the paper (sometimes it’s {0, 1}, other times {-1, 1})


**Time Spent Reviewing:**

7

---

> ### Author Response · Authors · 2021-08-10
> **Response to Reviewer 1 - comments on theory**
>
> We would like to thank the reviewer for the in-depth review and insightful comments, and for their support of the paper.
>
> **Comments on theory**
>
> > Theorem (that calibration works for sufficiently different training environments) is not surprising given IRM Theorem...
>
> While indeed some of our proof techniques bear a similarity to IRM, we believe that proving there is a strong connection between OOD generalization and (multi-domain) calibration is indeed novel. This connection, which to the best of our knowledge has not been highlighted before, opens the door to applying the large and constantly expanding body of research on **calibration** to the crucial field of **OOD generalization**.
>
> > More useful/convincing for their story would have been a theorem that shows where calibration works and IRM does not.
>
> This is an excellent point that we should have been clearer about in the paper. While we did not state this as a theorem, Figure 3(a) in the paper demonstrates a case where we show analytically that calibration captures invariance and IRMv1 does not.
> Generally, since both calibration over multiple domains and IRM seek the same type of conditional independence, the key question is how well does a proposed regularizer capture the sought conditional independence.
> The success of MMCE is a consequence of the consistency result provided in [1, Theorem 1], a type of guarantee that is not given for IRMv1. Theorems 2 and 3 in [1] are also directly applicable in the multiple-environment setting; they provide finite sample guarantees and approximation guarantees for ECE. Since these are straightforward applications of the results in [1], we decided to minimize the scope of this discussion in the main paper due to space constraints. That being said, we see the importance of stating these corollaries formally, while emphasizing the cases where the guarantees they provide come into effect and calibration based methods can provably outperform IRMv1. Hence, we will add a section to the supplementary material with an extended introduction of the MMCE penalty, the aforementioned corollaries and a full description of the case where these guarantees allow calibration based methods to outperform IRMv1.
>
> [1] Kumar, A., Sarawagi, S., & Jain, U. (2018, July). Trainable calibration measures for neural networks from kernel mean embeddings. In International Conference on Machine Learning (pp. 2805-2814). PMLR.

---

> ### Author Response · Authors · 2021-08-10
> **Response to Reviewer 1 - other comments and discussion on limitations**
>
> **Other comments and discussion on limitations**
>
> We greatly appreciate the detailed review. We will fix the typos, change the binary label in Thm. 1 to {0, 1} (the reason for choosing {-1, 1} was just for slightly cleaner notation), and expand the discussion on limitations to better reflect that further developments are required to fully understand where improved post-hoc calibration will not result in better OOD accuracy.

---

> ### Author Response · Authors · 2021-08-10
> **Response to Reviewer 1 - comments on methodological contribution**
>
> **Comments on methodological contribution**
>
> > the numbers in orig column for Camelyon 17 in Table 2 are not matching the WILDS...
>
> We thank the reviewer for raising this point, and agree that we should have been clearer about the modification made in the Camelyon17 experiments. Specifically, we used 4 environments split to training and validation sets, instead of 3 training environments and 1 validation environment, since in order to validate multi-domain calibration, the validation set must contain data from multiple environments. We will add a note to the appendix about the difference from the setting in the WILDS paper. We also added an anonymous code repository to provide reliable documentation of our settings and reproducible results: https://anonymous.4open.science/r/OOD_Calibration/
>
> When training with a smaller number of training environments, the best performing method is still CLOvE. As for other architectures, we could not finish all of the experiments in time for the author response deadline. In case of acceptance, we will include these results in the camera-ready version.
>
> > would be good to add a discussion about computational costs compared to other methods
>
> We agree that explicitly stating the computational costs associated with each method is beneficial. With CLOvE, the only consideration in terms of computational costs is due to the use of kernels in the objective function. This adds a computational cost that scales quadratically with the batch size, and is independent of the network size. Even with the relatively small network modules used for fine-tuning this was not a major concern; with large networks we find the added computational cost becomes negligible w.r.t the cost of the entire backprop operation.
> To estimate this numerically, we timed backprop iterations on a small multilayer perceptron. With a large batch size of 8192, a backprop iteration with CLOvE can be x1.5 longer than with IRMv1. However with medium batch sizes like 512 and 2048 which we use in our experiments, there is no statistically significant difference between the times per iteration.
>
> > need some more experimental details on how you use the clove regularizer in practice…
>
> Indeed as assumed by the reviewer, we use a similar setting to the code provided for [1]. This also appears in the anonymous github repository.
>
> > how have the authors chosen the parameters for measuring ECE and for post-hoc calibration (e.g. number of bins)?
>
> ECE is always measured with 10 bins, dividing the [0,1] interval into sub-intervals of equal length. This is specified in the code we provide. We leave exploring the ramifications of other binning strategies (e.g. the one proposed in [2]) to future work.
>
> > what if you don’t use Clove as fine-tuning but e.g. the IRMv1 regularizer on the entire network when trained from scratch? generally: are there scenarios in which post-hoc calibration fails to perform better than methods that exploit multi-domain information during training (e.g. GroupDRO, DeepCORAL, IRM)? moreover, when does a decrease in ECE not imply an improvement in OOD accuracy?
>
> We thank the reviewer for the useful, thought provoking comment. Following this comment, we explored potential failure modes and tried training with IRMv1 on the training and validation set without CLOvE. While results are still preliminary (i.e. fewer runs compared to the fine-tuning results), we find that training using the IRMv1 regularizer on the entire network when trained from scratch on both the training and validation underperforms CLOvE on average. That being said, we agree that exploring failure modes could shed light on the connection between OOD and post-hoc calibration, and this is something we plan to explore more thoroughly in the future.
> In the datasets that we experimented with we did not find cases where post-hoc calibration damaged accuracy significantly. However, we did see cases where it did not improve performance, e.g. when calibrating a model trained with IRM on CivilComments, as shown in Table 3 of the paper. Importantly, in our experiments we see that such failure to improve OOD performance coincided with a failure to improve ID average ECE.
>
> This is not to say that failure cases (i.e. where post-processing improves ID ECE but damages OOD accuracy) do not exist, just that across our experiments we did not encounter them, and that moderate improvements in ECE resulted in better OOD accuracy. An interesting point to note is that in our experiments post-hoc calibration did not damage ID accuracy, and one could hypothesize that failure cases will occur when post-hoc calibration decreases ID accuracy. This is somewhat related to the observations made in recent work regarding the correlation between ID accuracy and OOD accuracy [3], but is merely speculative and requires further research.
>
> [1] Kumar, A., Sarawagi, S., & Jain, U. (2018, July). Trainable calibration measures for neural networks from kernel mean embeddings. In International Conference on Machine Learning (pp. 2805-2814). PMLR.
>
> [2] Gupta, C., Podkopaev, A., & Ramdas, A. (2020). Distribution-free binary classification: prediction sets, confidence intervals and calibration. Advances in Neural Information Processing Systems, 33.
>
> [3] Miller, J. P., Taori, R., Raghunathan, A., Sagawa, S., Koh, P. W., Shankar, V., ... & Schmidt, L. (2021, July). Accuracy on the Line: On the Strong Correlation Between Out-of-Distribution and In-Distribution Generalization. In International Conference on Machine Learning (pp. 7721-7735). PMLR.

---

### Author Response · Authors · 2021-08-10
**Thanks for reviews, additional experiments and code repository**

We would like to thank the reviewers for their careful reading and thoughtful feedback. We also wish to acknowledge their support, and we are pleased they all found the paper clearly written, with “rich experiments” “demonstrating impressive effectiveness “ [R2, R1, R4], novel theoretical results [R2], with “important insights” [R3]. We are also glad the reviewers acknowledged the proposed connection between calibration and OOD generalization “opens up a path for further research w.r.t to theory and improved methods.”

Below we address in detail the comments and questions made individually by each of the reviewers - we look forward to an active discussion and being able to resolve any misunderstandings that might have arisen.
One important point raised by several reviewers is the fact we ran our experiments with fine-tuning the proposed loss, as opposed to end-to-end training. We note we did already train models end-to-end with our loss in smaller datasets like Colored MNIST. Following the reviewer comments, we further ran preliminary experiments with end-to-end training, and so far have found the results to be on par or worse than fine-tuning, and our proposed regularizer still outperforms the baselines. We do wish to note that fine-tuning is in general a common approach for improving model calibration, as well as a common practice when dealing with large NLP and computer vision models, which is why we considered it a suitable approach for our experiments in the first place.

We also added an anonymous code repository that can further clarify implementation details: https://anonymous.4open.science/r/OOD_Calibration/

---

### Decision · Program_Chairs · 2021-09-27

**Decision:**

Accept (Poster)

**Comment:**

Authors make substantive contributions to the connections between out-of-distribution generalization and calibration. I agree with the reviewers that the contributions are significant and interesting. To make the paper more accessible, I encourage the authors to make the exposition (at least the first few sections) more friendly to readers who don’t have a causal inference background.